# Decentralized TD Tracking with Linear Function Approximation and its Finite-Time Analysis

**Gang Wang**$^*$    **Songtao Lu**$^2$    **Georgios B. Giannakis**$^1$    **Gerald Tesauro**$^2$    **Jian Sun**$^3$

$^1$University of Minnesota, Minneapolis, MN 55455, US; `georgios@umn.edu`
$^2$IBM Research, Yorktown Heights, NY 10598, US; {`songtao@ibm.com, gtesauro@us.ibm.com`}
$^3$Beijing Institute of Technology, Beijing 100081, China; `sunjian@bit.edu.cn`

## Abstract

The present contribution deals with decentralized policy evaluation in multi-agent Markov decision processes using temporal-difference (TD) methods with linear function approximation for scalability. The agents cooperate to estimate the value function of such a process by observing continual state transitions of a shared environment over the graph of interconnected nodes (agents), along with locally private rewards. Different from existing consensus-type TD algorithms, the approach here develops a simple decentralized TD tracker by wedding TD learning with gradient tracking techniques. The non-asymptotic properties of the novel TD tracker are established for both independent and identically distributed (i.i.d.) as well as Markovian transitions through a unifying multistep Lyapunov analysis. In contrast to the prior art, the novel algorithm forgoes the limiting error bounds on the number of agents, which endows it with performance comparable to that of centralized TD methods that are the sharpest known to date.

## 1   Introduction

In reinforcement learning (RL), an agent relies on interactions with an environment to maximize a given cumulative reward. RL has made major advances in control and decision-making tasks encountered in areas as diverse as robotics, neuroscience, game theory, and artificial intelligence [35], [39]. It is particularly powerful when combined with deep neural network representations [27]. Temporal-difference (TD) learning [34] with gradient descent and function approximation is a cornerstone RL algorithm with documented success in several large-scale applications [40], [35], [31]. TD$^2$ learning offers a means of estimating the long-term cumulative future reward of a given policy as a function of the current state–what is referred to as policy evaluation. Function approximation is typically employed to estimate the expected cumulative future reward from a given state. Parameters of the function approximator are updated based on state transitions along with the associated rewards.

While TD updates are simple, a rigorous analysis of TD methods requires sophisticated tools. In this direction, a number of contributions have offered promising results in recent years. Asymptotic properties that hold as the number of updates grows to infinity (e.g., almost sure convergence, and convergence in mean) have been analyzed for TD methods with linear function approximation in e.g., [34, 10, 16, 12, 1, 41, 38, 36, 15]. To accommodate practical settings in control, signal processing, and machine learning tasks, where the data can be voluminous and continual [27, 35], research focus has gradually shifted toward non-asymptotic performance guarantees holding even with finite iterations, that is state transitions in the context of RL. Such analysis helps one understand the algorithm (a.k.a.

---

$^*$G. Wang's research in this paper was carried out when he was with the University of Minnesota. He is currently affiliated with Beijing Institute of Technology, China. (e-mail: `gangwang@bit.edu.cn`.)
$^2$Hereafter we use TD(0), abbreviated as TD.

agent)'s sample efficiency in terms of how many data are required in order to guarantee a desired level of solution accuracy.

A non-asymptotic (i.e., finite-time) analysis of TD learning with linear function approximation however, becomes more challenging than its asymptotic counterpart, because: i) consecutive state transitions along a Markov decision process (MDP) trajectory introduce correlation and bias in the corresponding TD updates, with respect to the limiting stationary distribution [3]; and ii) the TD updates do not correspond to minimizing any static objective as standard stochastic optimization algorithms do [25]. Despite these challenges, finite-time performance of TD methods (with linear function approximation) has recently been analyzed in [19, 24, 8, 20], and [18], assuming that the state transitions are i.i.d. Dealing with the more practical Markovian transitions, finite-sample error bounds are available for TD variants that include an additional projection step to control the gradient bias [3, 46], as well as for the original TD method [32, 43]. Finite-sample convergence properties of two-timescale and gradient TD generalizations have been also studied [9, 44, 47, 14].

Collectively, the above-mentioned efforts were made for single-agent RL, whereas algorithmic and theoretical developments in multi-agent RL, and policy evaluation in particular, still remain limited; see e.g., [7, 17, 26, 48, 42, 11, 33, 5]. In a cooperative multi-agent setting, a group of agents collaborate to learn the value function of a given policy based on locally private rewards observed from a shared environment, and information that is exchanged among neighbors [26, 42, 11, 33, 5]. Minimizing the mean-square projected Bellman error from finite i.i.d. transitions, distributed primal-dual methods have been developed for multi-agent policy evaluation using variance reduction [42, 5, 23, 22]. The first distributed TD learning algorithm for the online setting with continual data was reported in [26], and has been analyzed asymptotically using the ordinary differential equation (ODE) technique [4]. Finite-time bounds on the mean-square error of its iterates have recently been derived for i.i.d. data [11], and also for Markov correlated data [33]. However, optimality is not ensured in the sense that there is a gap between existing finite-time error bounds of distributed TD methods, and those of state-of-the-art decentralized stochastic optimization based ones in e.g., [30, 6, 45].

In this paper, we revisit the problem of multi-agent policy evaluation via decentralized TD learning. Aspiring to fill the gap, we develop a decentralized TD tracker (DTDT) with function approximation, by leveraging advances in variance reduction offered by gradient tracking. Moreover, we provide a unifying non-asymptotic analysis of decentralized TD methods with gradient tracking as well as linear function approximation, under i.i.d. and the practical yet challenging Markovian setting. The main theme of our analysis is on investigating the drift of an suitably chosen Lyapunov function to obtain bounds on the mean-square error of local parameter estimates and on the multi-agent consensus error. Our finite-time results establish that the novel decentralized TD tracker converges linearly to a neighborhood of the optimum under both i.i.d. and Markovian data. At least as important, the size of this neighborhood can be made arbitrarily small by selecting an appropriate stepsize that does not depend on the number of agents, unlike those achieved by existing decentralized TD methods.

**Notation.** Matrices (column vectors) are denoted by upper- (lower-) case boldface letters; sets by calligraphic letters; and the 2-norm of vectors by $\|\cdot\|$, which for matrices stands for the Frobenius norm. The $i$-the largest eigenvalue (singular value) of a matrix is denoted by $\lambda_i(\cdot)$ $(\sigma_i(\cdot))$.

## 2 Preliminaries

We begin by providing some standard background on MDPs, and briefly reviewing the centralized TD learning algorithm. More details on MDPs and RL in general can be referred to various sources; see e.g., [2], [35], and [37].

### 2.1 MDPs, value functions, and policy evaluation

An MDP is described by a 5-tuple $(\mathcal{S}, \mathcal{A}, p, r, \gamma)$, where $\mathcal{S} := \{s^1, s^2, \dots, s^{|\mathcal{S}|}\}$ denotes a finite state space, $\mathcal{A}$ represents a finite action space, $p = \{p(s'|s,a)\}_{s,s'\in\mathcal{S}, a\in\mathcal{A}}$ collects the probabilities $(p(s'|s,a) \geq 0)$ of transitioning to states $s'$ when taking action $a$ at current state $s$, with $\sum_{s'\in\mathcal{S}} p(s'|s,a) = 1$ for all $s \in \mathcal{S}$ and $a \in \mathcal{A}$, $r = \{r(s,a,s') \geq 0\}_{s,s'\in\mathcal{S}, a\in\mathcal{A}}$ is the reward corresponding to transition $(s,a,s')$ assumed uniformly upper bounded by $r_{\max} > 0$, and $\gamma \in [0,1)$ stands for the discount factor. The policy to be evaluated is a mapping $\pi : \mathcal{S} \times \mathcal{A} \to [0,1]$ determining the probability of taking action $a$ at state $s$. Starting from some $s_0 = s$, for all time $t \in \mathbb{N}$, let $a_t \sim \pi(s_t, \cdot)$, $s_{t+1} \sim p(\cdot|s_t, a_t)$, and $r_{t+1} = r(s_t, a_t, s_{t+1})$. Define also the average

reward $R^\pi(s) = \sum_{s' \in \mathcal{S}} \sum_{a \in \mathcal{A}} \pi(s, a)p(s'|s, a)r(s, a, s')$ for each $s$, and the average probability $P^\pi(s, s') = \sum_{a \in \mathcal{A}} \pi(s, a)p(s'|s, a)$ of transitioning from $s$ to $s'$.

The value function of a given policy $\pi$, denoted by $V^\pi : \mathcal{S} \to \mathbb{R}$, maps each state $s$ to a scalar measuring the expected discounted reward that will be received by following $\pi$ to take subsequent actions $\{a_t\}$ when starting from state $s_0 = s$; that is, $V^\pi(s) = \mathbb{E}_\pi[\sum_{t=0}^\infty \gamma^t r_{t+1} | s_0 = s]$. Here, the expectation $\mathbb{E}$ is taken over the MDP trajectory $(s_0, a_0, r_1, s_1, a_1, r_2, s_2, \ldots)$ generated by following $\pi$ from $s$. Without loss of generality, after renumbering entries of $\mathcal{S}$ as $[|\mathcal{S}|] := \{1, 2, \ldots, |\mathcal{S}|\}$, both functions $V^\pi(s)$ and $R^\pi(s)$ of $s \in \mathcal{S}$ can be viewed as vectors in $\mathbb{R}^{|\mathcal{S}|}$, and likewise, $\boldsymbol{P}^\pi$ as a matrix in $\mathbb{R}^{|\mathcal{S}| \times |\mathcal{S}|}$. It is known that $V^\pi$ satisfies the so-called Bellman equation, see e.g., [2]

$$V^\pi = R^\pi + \gamma P^\pi V^\pi. \tag{1}$$

Policy evaluation is the problem of finding $V^\pi$. If both $P^\pi$ and $R^\pi$ were known, $V^\pi$ can be computed exactly by inverting the Bellman equation (solving a linear system). In the context of learning however, the process dynamics is assumed unknown, and the goal is to estimate $V^\pi$ from observations gathered along a single continuous MDP trajectory with no resets instead. To simplify the notation, we will henceforth drop the superscript $(\cdot)^\pi$, since the policy to be evaluated is kept fixed.

## 2.2 TD learning with linear function approximation

In many real-world problems, the state space is often far too large to compute exact values of every state [2]. This 'curse of dimensionality' is typically addressed using function approximation methods. In particular, we focus here on a function approximation $V^\pi(s) \approx V_{\boldsymbol{\theta}}(s) = \boldsymbol{\phi}^\top(s)\boldsymbol{\theta}$ that is linear in a pre-selected feature vector $\boldsymbol{\phi}(s) \in \mathbb{R}^d$ of state $s$, where $\boldsymbol{\theta}$ is the unknown parameter vector to be learned. In practice, we have $d \ll |\mathcal{S}|$, which makes it possible to account for rarely visited or unvisited states. Per slot $t$, the states $s_t$ are invisible to the learning agent by any means other than via their corresponding $\boldsymbol{\phi}(s_t)$. Hence, this function approximation formulation includes partially observable (PO)MDPs as a special case. Now, the task boils down to estimating $\boldsymbol{\theta}$ such that $V_{\boldsymbol{\theta}} \approx V$.

The classical TD learning algorithm with linear function approximation [34, 35] starts with some guess $\boldsymbol{\theta}_0$ of the parameter vector. Upon observing the $t$-th transition $\zeta_{t+1} := (\boldsymbol{\phi}(s_t), r_t, \boldsymbol{\phi}(s_{t+1}))$, the so-called TD error $\delta(\boldsymbol{\theta}_t, \zeta_{t+1}) := r_t + \gamma V_{\boldsymbol{\theta}_t}(s_{t+1}) - V_{\boldsymbol{\theta}_t}(s_k)$ is found first, which is subsequently used to update parameter estimate $\boldsymbol{\theta}_t$ as

$$\boldsymbol{\theta}_{t+1} = \boldsymbol{\theta}_t + \alpha_t \boldsymbol{g}(\boldsymbol{\theta}_t, \zeta_{t+1}) = \boldsymbol{\theta}_t + \alpha_t \delta(\boldsymbol{\theta}_t, \zeta_{t+1})\nabla V_{\boldsymbol{\theta}_t}(s_t) \tag{2}$$

where $\alpha_t > 0$ is the stepsize, $\nabla V_{\boldsymbol{\theta}_t}(s_t)$ denotes the gradient of $V_{\boldsymbol{\theta}}(s_t)$ with respect to $\boldsymbol{\theta}$ evaluated at the current estimate $\boldsymbol{\theta}_t$, and the 'stochastic gradient' is given by

$$\boldsymbol{g}(\boldsymbol{\theta}_t, \zeta_{t+1}) := \boldsymbol{\phi}(s_t)[r_t + \gamma \boldsymbol{\phi}^\top(s_{t+1})\boldsymbol{\theta}_t - \boldsymbol{\phi}^\top(s_t)\boldsymbol{\theta}_t] \triangleq \boldsymbol{A}(\zeta_{t+1})\boldsymbol{\theta}_t + \boldsymbol{b}(\zeta_{t+1}) \tag{3}$$

where we have defined $\boldsymbol{A}(\zeta_{t+1}) := \boldsymbol{\phi}(s_t)[\gamma \boldsymbol{\phi}(s_{t+1}) - \boldsymbol{\phi}(s_t)]^\top$, and $\boldsymbol{b}(\zeta_{t+1}) := r_t \boldsymbol{\phi}(s_t)$ for brevity.

To proceed, we will need the following standard assumptions; see also [41], [3], [32], [8].

**Assumption 1** (*Ergodicity*). *The Markov chain $\{s_t\}_{t \geq 0}$ induced by policy $\pi$ is irreducible and aperiodic. There is a unique stationary distribution $\boldsymbol{\pi} \in \mathbb{R}^{|\mathcal{S}|}$ (with slight abuse of notation) such that*

$$\boldsymbol{\pi}^\top \boldsymbol{P} = \boldsymbol{\pi}^\top \tag{4}$$

*with $\pi(i) > 0$ for all $i \in [|\mathcal{S}|]$. Let $\mathbb{E}_{\boldsymbol{\pi}}[\cdot]$ stand for expectation with respect to distribution $\boldsymbol{\pi}$.*

**Assumption 2** (*Feature regularity*). *All features are bounded and linearly independent, i.e., $\|\boldsymbol{\phi}(i)\| \leq 1$ for all $i \in [|\mathcal{S}|]$, and the feature matrix $\boldsymbol{\Phi} := [\boldsymbol{\phi}(1)\, \boldsymbol{\phi}(2)\, \cdots\, \boldsymbol{\phi}(|\mathcal{S}|)]^\top \in \mathbb{R}^{|\mathcal{S}| \times d}$ is full rank.*

Under As. 1—2 and introducing the diagonal matrix $\boldsymbol{D} := \mathrm{diag}([\pi(1)\, \pi(2)\, \cdots\, \pi(|\mathcal{S}|)]) \in \mathbb{R}^{|\mathcal{S}| \times |\mathcal{S}|}$, it can be shown that the following limits hold true (e.g., [41])

$$\boldsymbol{A} := \lim_{t \to \infty} \mathbb{E}[\boldsymbol{A}(\zeta_t)] = \boldsymbol{\Phi}^\top \boldsymbol{D}(\gamma \boldsymbol{P}\boldsymbol{\Phi} - \boldsymbol{\Phi}) \prec \boldsymbol{0}, \tag{5a}$$

$$\boldsymbol{b} := \lim_{t \to \infty} \mathbb{E}[\boldsymbol{b}(\zeta_t)] = \boldsymbol{\Phi}^\top \boldsymbol{D}R, \tag{5b}$$

which implies that the sequence of stochastic gradients $\{\boldsymbol{g}(\boldsymbol{\theta}, \zeta_t)\}$ converges in mean to $\boldsymbol{g}(\boldsymbol{\theta}) := \boldsymbol{A}\boldsymbol{\theta} + \boldsymbol{b}$ obtained at the stationary distribution. It has been shown in [41, Thm. 1] that the sequence of TD parameter iterates generated by (2) with appropriate stepsizes converges as $t \to \infty$ to the fixed point $\boldsymbol{\theta}^* = -\boldsymbol{A}^{-1}\boldsymbol{b}$, so that $\boldsymbol{g}(\boldsymbol{\theta}^*) = \boldsymbol{0}$. However, along the trajectory of a Markov chain starting from an arbitrary distribution, transitions collected 'on-the-fly' render stochastic gradients $\boldsymbol{g}(\boldsymbol{\theta}, \zeta_t)$ biased from $\boldsymbol{g}(\boldsymbol{\theta})$ as well as correlated with $\boldsymbol{g}(\boldsymbol{\theta}, \zeta_{t-1})$ and $\boldsymbol{g}(\boldsymbol{\theta}, \zeta_{t+1})$. This is indeed the major hurdle that has challenged non-asymptotic analysis of RL algorithms until recently.

## 3 Multi-agent MDPs and decentralized TD tracking

### 3.1 Multi-agent MDPs

The goal of this work is to deal with decentralized policy evaluation for multi-agent (MA) RL, where several agents cooperate to compute the value function in a shared environment. Consider a set $\mathcal{N}$ of agents with $|\mathcal{N}| = N$, distributed over a communication network $\mathcal{G} = (\mathcal{N}, \mathcal{E})$, where $\mathcal{E} \subseteq \mathcal{N} \times \mathcal{N}$ is the edge set. For $n \in \mathcal{N}$, let $\mathcal{N}_n \subseteq \mathcal{N}$ denote the set of neighbor(s) of agent $n$ between which simple information exchanges are possible. We consider each agent implements a local policy $\pi^n$ that can be different from other policies. As elaborated in the single-agent setting, when combined with the joint policy $\pi := \{\pi^n\}_{n \in \mathcal{N}}$, a multi-agent MDP can be described by the following 6-tuple

$$\left( \mathcal{S}, \{\mathcal{A}^n\}_{n=1}^N, p, \{r^n\}_{n=1}^N, \gamma, \mathcal{G} \right) \tag{6}$$

where $\mathcal{S}$ is a finite set of states shared by all agents, $\mathcal{A}^n$ is a finite set of actions available to agent $n$, $p := \{p(s'|s, \boldsymbol{a})\}_{s,s' \in \mathcal{S}, \boldsymbol{a} \in \mathcal{A}}$ with $\boldsymbol{a} := (a^1, a^2, \dots, a^n) \in \mathcal{A}^1 \times \mathcal{A}^2 \times \cdots \times \mathcal{A}^n \triangleq \mathcal{A}$ being the joint action, and $r^n := \{0 \leq r^n(s, \boldsymbol{a}, s') \leq r_{\max}\}_{s,s' \in \mathcal{S}, \boldsymbol{a} \in \mathcal{A}}$ is the reward function for agent $n$. It is worth remarking that there is no centralized controller that can observe all information exchanged; instead, each agent can observe the state $s \in \mathcal{S}$ of the shared environment, whereas its action $a^n \in \mathcal{A}^n$ and rewards $r^n(s, \boldsymbol{a}, s')$ are kept private from others. Likewise, we introduce per agent $n$ the average reward $R^n(s) = \sum_{s' \in \mathcal{S}} \sum_{\boldsymbol{a} \in \mathcal{A}} \pi(s, \boldsymbol{a}) p(s'|s, \boldsymbol{a}) r^n(s, \boldsymbol{a}, s')$ for each $s$, and also the average probability $P(s, s') = \sum_{\boldsymbol{a} \in \mathcal{A}} \pi(s, \boldsymbol{a}) p(s'|s, \boldsymbol{a})$ of transitioning from state $s$ to $s'$.

Specifically at time $t$, each agent $n$ observes the current state $s_t \in \mathcal{S}$ and chooses an action $a^n \in \mathcal{A}^n$ according to policy $\pi^n$. Based on the joint action of all agents, the environment moves to $s_{t+1}$, for which an immediate reward $r_t^n := r^n(s_t, \boldsymbol{a}_{t+1}, s_{t+1})$ is revealed to agent $n$. The goal of multi-agent policy evaluation is to collaboratively compute the average of the expected sums of discounted rewards from a network of agents; that is,

$$V_{\mathcal{G}}(s) = \mathbb{E}_\pi \left[ \frac{1}{N} \sum_{n \in \mathcal{N}} \sum_{t=0}^\infty \gamma^{t+1} r_t^n \big| s_0 = s \right]. \tag{7}$$

Replacing $R^\pi$ in (1) with the average over a network of agents, it can be shown that $V_{\mathcal{G}}$ obeys

$$V_{\mathcal{G}} = N^{-1} \sum_{n \in \mathcal{N}} R^n + \gamma \boldsymbol{P} V_{\mathcal{G}}. \tag{8}$$

The curse of dimensionality associated with exactly computing $V_{\mathcal{G}}$ when $|\mathcal{S}|$ is large, prompts us to pursue a linear approximation as $V_{\mathcal{G}}(s) \approx V_{\boldsymbol{\theta}}(s) = \boldsymbol{\phi}^\top(s) \boldsymbol{\theta}$, parameterized by an unknown $\boldsymbol{\theta} \in \mathbb{R}^d$ with $d \ll |\mathcal{S}|$. The goal of the agents is to cooperatively find $\boldsymbol{\theta}$ such that $V_{\mathcal{G}} \approx V_{\bar{\boldsymbol{\theta}}}$ based on observations collected along a single trajectory of a multi-agent MDP denoted by $(s_0, \{a_0^n\}_{n \in \mathcal{N}}, \{r_1^n\}_{n \in \mathcal{N}}, s_1, \{a_1^n\}_{n \in \mathcal{N}}, \{r_2^n\}_{n \in \mathcal{N}}, s_2, \dots)$ with actions and rewards kept private, meaning without a central controller observing the entire trajectory. To tackle this problem, decentralized variants of TD methods have been developed and analyzed in [26], [11], [33]. However, their bounds are $N$ times worse than that of centralized TD methods [32, 43], because local gradient estimators have high variance. The reason is regarding the partial observability, where each agent updates its own parameter via the local TD error. In this work, our fresh idea is to leverage local gradient trackers to approximate the global gradient found as if there is a central controller observing all information, and perform decentralized TD updates, which justifies the name of our algorithm—*decentralized TD tracking* (DTDT).

### 3.2 Decentralized TD tracking with linear function approximation

Decentralized gradient tracking methods build on the intuitive idea that, if every agent could have access to the global gradient estimate $N^{-1} \sum_{n \in \mathcal{N}} \boldsymbol{g}(\boldsymbol{\theta}_t^n, \zeta_t^n)$ per slot $t \geq 0$, then the centralized TD updates can be implemented at each agent. The gradient tracking technique offers a simple yet effective means of doing so approximately. To develop our DTDT algorithm, let us first introduce an auxiliary variable $\boldsymbol{\psi}^n$ per agent $n \in \mathcal{N}$, dedicated to tracking the global gradient locally, and obtained by mixing the estimates of its neighbors as well as refreshing its local one.

Specifically, upon receiving the parameter estimates $\{\boldsymbol{\theta}_t^n\}$ and the gradient trackers $\{\boldsymbol{\psi}_t^n\}$ from its neighbors $n \in \mathcal{N}$, each agent $n$ performs two steps: s1) updates the mix of all available parameter estimates $\{\boldsymbol{\theta}_t^n\}_{n \in \mathcal{N}}$ using $\boldsymbol{\psi}_t^n$; and, s2) refines the mix of all available gradient trackers $\{\boldsymbol{\psi}_t^n\}_{n \in \mathcal{N}}$ with the local gradient $\boldsymbol{g}(\boldsymbol{\theta}_{t+1}^n, \zeta_{t+1}^n) = \delta(\boldsymbol{\theta}_{t+1}^n, \zeta_{t+1}^n) \boldsymbol{\phi}(s_t)$ (cf. (3)) corresponding to the new transition $\zeta_{t+1}^n := (\boldsymbol{\phi}(s_t), r_t^n, \boldsymbol{\phi}(s_{t+1}))$, yielding the DTDT recursions for all $i \in \mathcal{N}$ and $t \geq 0$:

$$\boldsymbol{\theta}_{t+1}^n = \sum_{n' \in \mathcal{N}_n} W_{nn'} \boldsymbol{\theta}_t^{n'} + \alpha_t \boldsymbol{\psi}_t^n \tag{9a}$$

---

**Algorithm 1** Decentralized TD Tracking (DTDT) with linear function approximation

---
1: **Input:** stepsize $\alpha_t > 0$, features $\{\phi(s)\}_{s\in\mathcal{S}}$, and weight matrix $\boldsymbol{W}$.
2: **Initialize:** $\{\boldsymbol{\theta}_0^n = \boldsymbol{\theta}_0\}_{n\in\mathcal{N}}$, $\{\boldsymbol{\psi}_0^n = \boldsymbol{0}\}_{n\in\mathcal{N}}$, and $\{\boldsymbol{g}(\boldsymbol{\theta}_0^n, \zeta_0^n) = \boldsymbol{0}\}_{n\in\mathcal{N}}$.
3: **for** $t = 0, 1, \cdots, T$ **do**
4:     **for** $n = 1, 2, \cdots, N$ **do**                                                    ▷ Computation over a graph
5:         Agent $n$ receives $\boldsymbol{\theta}_t^{n'}$ and $\boldsymbol{\psi}_t^{n'}$ from its neighbors $n' \in \mathcal{N}_n$;
6:         Agent $n$ obtains $\boldsymbol{\theta}_{t+1}^n$ according to (9a);
7:         Agent $n$ observes $\zeta_{t+1}^n = (\phi(s_t), r_t^n, \phi(s_{t+1}))$, and computes $\boldsymbol{g}(\boldsymbol{\theta}_{t+1}^n, \zeta_{t+1}^n)$ via (3);
8:         Agent $n$ updates $\boldsymbol{\psi}_{t+1}^n$ according to (9b).
9:     **end for**
10: **end for**

---

$$\boldsymbol{\psi}_{t+1}^n = \sum_{n'\in\mathcal{N}_n} W_{nn'}\boldsymbol{\psi}_t^{n'} + \boldsymbol{g}(\boldsymbol{\theta}_{t+1}^n, \zeta_{t+1}^n) - \boldsymbol{g}(\boldsymbol{\theta}_t^n, \zeta_t^n) \tag{9b}$$

where, evidently, $\boldsymbol{\psi}_t^n$ tracks the global gradient $N^{-1}\sum_{n=1}^N \boldsymbol{g}(\boldsymbol{\theta}_t^n, \zeta_t^n)$ locally; and $W_{nn'}$ is a weight attached to the edge $(n, n')$ satisfying $W_{nn'} > 0$ if $n' \in \mathcal{N}_n$, and $W_{nn'} = 0$, otherwise.

To proceed, let us define $\boldsymbol{A}(\zeta_{t+1}) := \phi(s_t)[\gamma\phi^\top(s_{t+1}) - \phi^\top(s_t)]$ with $\zeta_{t+1} := (s_t, \{r_t^n\}_{n\in\mathcal{N}}, s_{t+1})$; and $\boldsymbol{b}(\zeta_{t+1}^n) := r_t^n\phi(s_t)$. Furthermore, we stack up all local parameter estimates $\{\boldsymbol{\theta}_t^n\}_{n\in\mathcal{N}}$, gradient trackers $\{\boldsymbol{\psi}_t^n\}_{n\in\mathcal{N}}$, and immediate rewards $\{r_t^n\}_{n\in\mathcal{N}}$ into matrices

$$\boldsymbol{\Theta}_t := [\boldsymbol{\theta}_t^1\ \boldsymbol{\theta}_t^2\ \cdots\ \boldsymbol{\theta}_t^N]^\top, \quad \boldsymbol{\Psi}_t := [\boldsymbol{\psi}_t^1\ \boldsymbol{\psi}_t^2\ \cdots\ \boldsymbol{\psi}_t^N]^\top, \quad \text{and} \quad \boldsymbol{r}_t := [r_t^1\ r_t^2\ \cdots\ r_t^N]^\top \tag{10}$$

and likewise for all local gradients

$$\boldsymbol{G}(\boldsymbol{\Theta}_t, \zeta_t) := [\boldsymbol{g}(\boldsymbol{\theta}_t^1, \zeta_t^1)\ \boldsymbol{g}(\boldsymbol{\theta}_t^2, \zeta_t^2)\ \cdots\ \boldsymbol{g}(\boldsymbol{\theta}_t^N, \zeta_t^N)]^\top = \boldsymbol{\Theta}_t\boldsymbol{A}^\top(\zeta_t) + \boldsymbol{r}_{t-1}\phi^\top(s_{t-1}). \tag{11}$$

With these definitions, the proposed multi-agent, decentralized TD tracking (DTDT) algorithm with linear function approximation in (9) can be compactly re-written as

$$\boldsymbol{\Theta}_{t+1} = \boldsymbol{W}\boldsymbol{\Theta}_t + \alpha_t\boldsymbol{\Psi}_t \tag{12a}$$

$$\boldsymbol{\Psi}_{t+1} = \boldsymbol{W}\boldsymbol{\Psi}_t + \boldsymbol{G}(\boldsymbol{\Theta}_{t+1}, \zeta_{t+1}) - \boldsymbol{G}(\boldsymbol{\Theta}_t, \zeta_t). \tag{12b}$$

Our DTDT algorithm with linear function approximation is tabulated as Alg. 1

### 3.3 Fixed-point characterization of decentralized TD tracking

We make the following standard assumptions that will be needed for our performance analysis.

**Assumption 3.** *The graph $\mathcal{G}$ corresponding to the network of agents is undirected and connected.*

**Assumption 4.** *The mixing matrix $\boldsymbol{W} := [W_{nn'} \geq 0, n, n' \in \mathcal{N}]$ is doubly stochastic, that is, $\boldsymbol{W}\boldsymbol{1} = \boldsymbol{1}$ and $\boldsymbol{1}^\top\boldsymbol{W} = \boldsymbol{1}^\top$, where $\boldsymbol{1}$ denotes all-one vectors of suitable dimensions.*

Let us define the following quantities averaged over the entire network of agents

$$\bar{\boldsymbol{\theta}}_t := N^{-1}\boldsymbol{\Theta}_t^\top\boldsymbol{1}, \ \ \bar{\boldsymbol{\psi}}_t := N^{-1}\boldsymbol{\Psi}_t^\top\boldsymbol{1}, \ \text{ and } \ \bar{\boldsymbol{g}}(\bar{\boldsymbol{\theta}}_t, \zeta_t) := N^{-1}\boldsymbol{G}^\top(\boldsymbol{\Theta}_t, \zeta_t)\boldsymbol{1} = \boldsymbol{A}(\zeta_t)\bar{\boldsymbol{\theta}}_t + \bar{r}_{t-1}\phi(s_{t-1}).$$

When initializing $\boldsymbol{\Psi}_0 = \boldsymbol{0}$ and $\boldsymbol{G}(\boldsymbol{\Theta}_0, \zeta_0) = \boldsymbol{0}$, it can be readily shown that (cf. (12))

$$\bar{\boldsymbol{\theta}}_{t+1} = \bar{\boldsymbol{\theta}}_t + \alpha_t\bar{\boldsymbol{\psi}}_t \tag{13a}$$

$$\bar{\boldsymbol{\psi}}_{t+1} = \bar{\boldsymbol{g}}(\bar{\boldsymbol{\theta}}_{t+1}, \zeta_{t+1}) = \boldsymbol{A}(\zeta_{t+1})\bar{\boldsymbol{\theta}}_{t+1} + \bar{r}_t\phi(s_t) \tag{13b}$$

yielding the parameter average system

$$\bar{\boldsymbol{\theta}}_{t+1} = \bar{\boldsymbol{\theta}}_t + \alpha_t[\boldsymbol{A}(\zeta_t)\bar{\boldsymbol{\theta}}_t + \bar{r}_{t-1}\phi(s_{t-1})] \tag{14}$$

which resembles the single-agent TD update in 2. Similar to (5b), define $\boldsymbol{b}^n := \lim_{t\to\infty}\mathbb{E}[\boldsymbol{b}(\zeta_t^n)]$, and $\bar{\boldsymbol{b}} := N^{-1}\sum_{n\in\mathcal{N}}\boldsymbol{b}^n$, so for any given $\bar{\boldsymbol{\theta}}$, we have $\bar{\boldsymbol{g}}(\bar{\boldsymbol{\theta}}) = \lim_{t\to\infty}\mathbb{E}[\bar{\boldsymbol{g}}(\bar{\boldsymbol{\theta}}, \zeta_t)] = \boldsymbol{A}\bar{\boldsymbol{\theta}} + \bar{\boldsymbol{b}}$.

Under As. 1 and 2 along with minimal conditions on $\alpha_t > 0$, [41, Thm. 1] asserts that Alg. 1 also converges asymptotically to the following fixed point (obtained by setting $\bar{\boldsymbol{g}}(\bar{\boldsymbol{\theta}}) = \boldsymbol{0}$)

$$\bar{\boldsymbol{\theta}}^* = -\boldsymbol{A}^{-1}\bar{\boldsymbol{b}}. \tag{15}$$

# 4 Finite-time analysis of decentralized TD tracking

Although gradient tracking is known to help reduce the variance of stochastic gradients, and overall improves the convergence in decentralized optimization [30, 45], it remains unclear whether and to what extent gradient tracking can benefit decentralized TD learning algorithms. Seeking to explore this direction, a non-asymptotic analysis of Alg. 1 is well motivated. This is the theme of this section that will develop a unifying finite-time analysis for our DTDT algorithm in the case of constant stepsizes ($\alpha_t = \alpha$) when observed data are i.i.d. or Markov correlated.

In our analysis, we will rely on the following results that hold regardless of the observation model.

**Lemma 1.** *If As. 3 and 4 are satisfied, and $\eta$ is the spectral radius of matrix $\boldsymbol{W} - N^{-1}\mathbf{1}\mathbf{1}^\top$, it holds for $\eta < 1$ and for all $\boldsymbol{\Theta} \in \mathbb{R}^{N \times p}$ that*

$$\|\boldsymbol{W}\boldsymbol{\Theta} - \mathbf{1}\bar{\boldsymbol{\theta}}^\top\| \leq \eta\|\boldsymbol{\Theta} - \mathbf{1}\bar{\boldsymbol{\theta}}^\top\|. \tag{16}$$

**Lemma 2** ($\boldsymbol{\Theta}$-iterate contraction)**.** *Under As. 3 and 4, it holds for all iterates $\{\boldsymbol{\Theta}_t\}_t$ that*

$$\|\boldsymbol{\Theta}_{t+1} - \mathbf{1}\bar{\boldsymbol{\theta}}_{t+1}^\top\|^2 \leq (1+\tau)\eta\|\boldsymbol{\Theta}_t - \mathbf{1}\bar{\boldsymbol{\theta}}_t^\top\|^2 + \alpha^2(1+1/\tau)\|\boldsymbol{\Psi}_t - \mathbf{1}\bar{\boldsymbol{\psi}}_t^\top\|^2 \tag{17}$$

*where the constant $\tau > 0$ is selected such that $(1+\tau)\eta < 1$.*

**Lemma 3** ($\boldsymbol{\Psi}$-iterate contraction)**.** *Under As. 2— 4, the next contraction holds for all iterates $\{\boldsymbol{\Psi}_t\}_t$*

$$\begin{aligned}
\|\boldsymbol{\Psi}_{t+1} - \mathbf{1}\bar{\boldsymbol{\psi}}_{t+1}^\top\|^2 \leq &(1+\tau)\eta^2\|\boldsymbol{\Psi}_t - \mathbf{1}\bar{\boldsymbol{\psi}}_t^\top\|^2 + 24(1+1/\tau)(1+\gamma)^2\|\boldsymbol{\Theta}_t - \mathbf{1}\bar{\boldsymbol{\theta}}_t^\top\|^2 \\
&+ 24N(1+1/\tau)(1+\gamma)^2\|\bar{\boldsymbol{\theta}}_{t+1} - \bar{\boldsymbol{\theta}}^*\|^2 + 120N(1+1/\tau)(1+\gamma)^2\|\bar{\boldsymbol{\theta}}_t - \bar{\boldsymbol{\theta}}^*\|^2 \\
&+ 96N(1+1/\tau)(1+\gamma)^2\|\bar{\boldsymbol{\theta}}^*\|^2 + 6N(1+1/\tau)r_{\max}^2.
\end{aligned} \tag{18}$$

Proofs of Lemmas 1—3 are provided in Appendices C—E of the supplementary material, respectively.

## 4.1 I.I.D. data

We begin by analyzing the non-asymptotic properties of DTDT when transitions $\zeta_t = (\phi(s_t), \{r_t^n\}_n, \phi(s_{t+1}))$ are i.i.d. In practice, it is hard to obtain i.i.d. data as pointed out in [8], yet the i.i.d. setting is discussed here for completeness of our theoretical developments.

**Lemma 4** ($\bar{\boldsymbol{\theta}}$-iterate contraction in IID setting)**.** *Under As. 2— 4, it holds for all iterates $\{\bar{\boldsymbol{\theta}}_t\}_t$ that*

$$\begin{aligned}
\mathbb{E}\big[\|\bar{\boldsymbol{\theta}}_{t+1} - \bar{\boldsymbol{\theta}}^*\|^2\big] \leq &\big[1 + 2\alpha\lambda_1 + 4\alpha^2(1+\gamma)^2\big]\mathbb{E}\big[\|\bar{\boldsymbol{\theta}}_t - \bar{\boldsymbol{\theta}}^*\|^2\big] + \frac{4\alpha^2(1+\gamma)^2}{N^2}\mathbb{E}\big[\|\boldsymbol{\Theta}_t - \mathbf{1}\bar{\boldsymbol{\theta}}_t^\top\|^2\big] \\
&+ 4\alpha^2(1+\gamma)^2\|\bar{\boldsymbol{\theta}}^*\|^2 + 4\alpha^2 r_{\max}^2
\end{aligned} \tag{19}$$

*where $\lambda_1 := \lambda_1(\boldsymbol{A}) < 0$ is the largest eigenvalue of the negative definite matrix $\boldsymbol{A}$ (cf. (5a)), and $\alpha > 0$ is chosen such that $0 < 1 + 2\alpha\lambda_1 < 1$.*

The proof of Lemma 4 can be found in Appendix F of the supplementary material. With the three iterate-contraction properties summarized in Lemmas 2—4, we are in a position to present our novel finite-time error bounds for DTDT when transitions are i.i.d..

**Theorem 1.** *Let As. 3 and 4 hold and take $\tau = (1-\eta)/(2\eta) > 0$. Then, there exist constant $\bar{\alpha}_i$ such that for any stepsize satisfying $0 < \alpha < \bar{\alpha}_i$, both the estimation error $\mathbb{E}[\|\bar{\boldsymbol{\theta}}_t - \bar{\boldsymbol{\theta}}^*\|^2]$ and the consensus disagreement $\mathbb{E}[\|\boldsymbol{\Theta}_{t+1} - \mathbf{1}\bar{\boldsymbol{\theta}}_t^\top\|^2]$ of Alg. 1 converge linearly with rate $\rho_i(\alpha) < 1$ to a neighborhood of the fixed-point; that is,*

$$\max\Big\{\mathbb{E}\big[\|\bar{\boldsymbol{\theta}}_t - \bar{\boldsymbol{\theta}}^*\|^2\big],\ N^{-1}\mathbb{E}\big[\|\boldsymbol{\Theta}_t - \mathbf{1}\bar{\boldsymbol{\theta}}_t^\top\|^2\big]\Big\} \leq c_0[\rho_i(\alpha)]^t + c_1(\bar{\alpha}_i)\alpha, \quad \forall t \geq 1 \tag{20}$$

*where $c_1(\bar{\alpha}_i) := \frac{1}{1-\rho_i(\bar{\alpha}_i)}\Big[\frac{4|\lambda_1| + 20\alpha(1+\gamma)^2}{5 - \bar{\alpha}_i|\lambda_1|}\|\bar{\boldsymbol{\theta}}^*\|^2 + \frac{|\lambda_1|r_{\max}^2}{4(1+\gamma)^2(5 - \bar{\alpha}_i|\lambda_1|)} + \frac{10r_{\max}^2\bar{\alpha}_i}{5 - \bar{\alpha}_i|\lambda_1|}\Big]$, $c_0 := \|\boldsymbol{\theta}_0 - \bar{\boldsymbol{\theta}}^*\|^2$, with the explicit expressions of $\bar{\alpha}_i$ and $\rho_i(\alpha)$ presented in Appendix A.*

**Remark 1.** *Let $N^{-1}\mathbb{E}[\|\boldsymbol{\Theta}_t - \mathbf{1}(\bar{\boldsymbol{\theta}}^*)^\top\|^2]$ measure the average quality of solutions obtained by all agents. In light of (20), we have that*

$$N^{-1}\mathbb{E}[\|\boldsymbol{\Theta}_t - \mathbf{1}(\bar{\boldsymbol{\theta}}^*)^\top\|^2] \leq 2N^{-1}\mathbb{E}\big[\|\boldsymbol{\Theta}_t - \mathbf{1}\bar{\boldsymbol{\theta}}_t^\top\|^2\big] + 2\mathbb{E}\big[\|\bar{\boldsymbol{\theta}}_t - \bar{\boldsymbol{\theta}}^*\|^2\big] \leq 4c_0[\rho_i(\alpha)]^t + 4c_1\alpha \tag{21}$$

*where the residual error term $4c_1\alpha$ does not depend on the network size $N$, and improves upon those reported in [11, Thm. 2] and [33, Prop. 1] by a factor of $1/N$, thanks to the gradient tracking.*

## 4.2 Markovian data

Thus far, performance of the proposed DTDT algorithm has been analyzed for the ideal setup of i.i.d. data. A more practical yet challenging case pertains to the Markovian transitions $\{\phi(s_t), \{r_t^n\}_{n\in\mathbb{N}}, \phi(s_{t+1})\}_{t\geq 0}$ gathered along a single trajectory of the multi-agent MDP. In contrast with i.i.d. data, Markovian data render consecutive TD updates correlated, and hence incur sizable gradient bias. Fortunately, any finite-state, irreducible, and aperiodic Markov chain converges to its unique stationary distribution geometrically fast [21, Thm. 4.9]. In light of this property, we establish the following result whose proof can be found in Appendix G of the supplementary material.

**Lemma 5** (Geometric ergodicity). *Under As. 1—4, the following holds for each $\mathbf{\Theta} \in \mathbb{R}^{N\times p}$*

$$\left\| \frac{1}{TN} \sum_{k=t}^{t+T-1} \mathbb{E}\big[ \boldsymbol{G}^\top(\mathbf{\Theta}, \zeta_k)\mathbf{1}|\zeta_0 \big] - \bar{\boldsymbol{g}}(\bar{\boldsymbol{\theta}}) \right\| \leq \sigma(T;t)\big(\|\bar{\boldsymbol{\theta}} - \bar{\boldsymbol{\theta}}^*\| + 1\big), \quad \forall t \in \mathbb{N}^+ \qquad (22)$$

*where $\sigma(T;t) := \frac{(1+\gamma)\nu_0\rho^t}{(1-\rho)T} \max\{2\|\bar{\boldsymbol{\theta}}^*\| + r_{\max}, 1\}$, with $\nu_0 > 0$ and $0 < \rho < 1$ are constants.*

Lemma 5 implies that the bias of the gradient average over $N$ agents and $T$ future slots, diminishes geometrically fast, thus offering a possible means for dealing with biased gradients arising from the Markovian data. Building on this observation, we are prompted to consider a multistep Lyapunov function that lends itself to effect a $\bar{\boldsymbol{\theta}}$-iterate contraction for DTDT from the Markovian data as follows $\mathcal{V}_t(T) = \frac{1}{2}\sum_{k=t}^{t+T-1} \|\bar{\boldsymbol{\theta}}_t - \bar{\boldsymbol{\theta}}^*\|^2$, where parameter $T \geq 1$ is chosen so that gradient bias can be controlled to yield contracting $\bar{\boldsymbol{\theta}}$-iterates as follows.

**Lemma 6** ($\bar{\boldsymbol{\theta}}$-iterate contraction in the Markovian setting). *Let $k_1 := (1+\gamma)^2$ and $k_2 := (1+\gamma)^2[3 + T\alpha'(1+\gamma)]$ with constant $0 \leq \alpha' \leq 1$. Under As. 2— 4, it holds for all $\{\bar{\boldsymbol{\theta}}_{t+T}\}_{t\geq 1}$ that*

$$\mathbb{E}\big[\|\bar{\boldsymbol{\theta}}_{t+T} - \bar{\boldsymbol{\theta}}^*\|^2\big] \leq \frac{9N + 18\alpha NT[\lambda_1 + 2\sigma(T;t)] + \alpha^2 T^2(72k_1 + 9Nk_2) + 2\alpha^4 T^4 k_2^2}{9N} \mathbb{E}\big[\|\bar{\boldsymbol{\theta}}_t - \bar{\boldsymbol{\theta}}^*\|^2\big]$$

$$+ \frac{3\alpha^2 T^2(48k_1 + Nk_2) + 4\alpha^4 T^4 k_2^2}{18N^2} \mathbb{E}\big[\|\mathbf{\Theta}_t - \mathbf{1}\bar{\boldsymbol{\theta}}_t^\top\|^2\big] + \frac{3\alpha^2 T^2(48k_1 + Nk_2) + 4\alpha^4 T^4 k_2^2}{18N} \|\bar{\boldsymbol{\theta}}^*\|^2$$

$$+ \frac{3\alpha^2 T^2(k_2 + 48) + 4\alpha^4 T^4 k_2^2}{18} r_{\max}^2 + \frac{\alpha T\sigma(T;t)}{2} \qquad (23)$$

*where the pair $(\alpha, T) > 0$ is chosen so that $0 < 9N + 18\alpha NT[\lambda_1 + 2\sigma(T;t)] + \alpha^2 T^2(72k_1 + 9Nk_2) + 2\alpha^4 T^4 k_2^2 < 1$, by leveraging that $\lambda_1 < 0$, and $\sigma(T,t)$ vanishes exponentially.*

The proof of Lemma 6 is postponed to Appendix D of the supplementary material. Indeed, this result is central to our finite-sample analysis of the proposed DTDT algorithm dealing with Markovian data, that is summarized next.

**Theorem 2.** *Under As. 2—4, fixing any stepsize $0 < \alpha < \bar{\alpha}_{\mathrm{m}} = \mathcal{O}(1)$, and taking $T = T_0 = \max\big\{ \frac{4\nu_0(1+\gamma)\max\{2\|\bar{\boldsymbol{\theta}}^*\| + r_{\max}, 1\}}{(1-\rho)|\lambda_1|}, \frac{288(1+1/\tau)(1+\gamma)^2}{|\lambda_1|} \big\}$, ensure that estimation error $\mathbb{E}[\|\bar{\boldsymbol{\theta}}_t - \bar{\boldsymbol{\theta}}^*\|^2]$ and consensus error $\mathbb{E}[\|\mathbf{\Theta}_t - \mathbf{1}\bar{\boldsymbol{\theta}}_t^\top\|^2]$ both converge linearly with rate $\rho_{\mathrm{m}}(\alpha, T_0) \in (0,1)$; that is,*

$$\max\Big\{ \mathbb{E}\big[\|\bar{\boldsymbol{\theta}}_t - \bar{\boldsymbol{\theta}}^*\|^2\big], N^{-1}\mathbb{E}\big[\|\mathbf{\Theta}_t - \mathbf{1}\bar{\boldsymbol{\theta}}_t^\top\|^2\big] \Big\} \leq c_2(\bar{\alpha}_{\mathrm{m}}, T_0)[\rho_{\mathrm{m}}(\alpha, T_0)]^t + c_3(\bar{\alpha}_{\mathrm{m}}, T_0)\alpha, \ \ \forall t \geq 1 \qquad (24)$$

*where $c_2(\bar{\alpha}_{\mathrm{m}}, T_0) > 0$, and $c_3(\bar{\alpha}_{\mathrm{m}}, T_0) > 0$ are appropriate constants depending on $T_0$ and $\bar{\alpha}_{\mathrm{m}}$, but not on $t \geq 0$ and $N$.*

Exact forms of constants $\bar{\alpha}_{\mathrm{m}}$, $\rho_{\mathrm{m}}(\alpha, T_0)$, $c_2(\bar{\alpha}_{\mathrm{m}}, T_0)$, and $c_3(\bar{\alpha}_{\mathrm{m}}, T_0)$ can be found in the proof of Thm. 2 provided in Appendix B of the supplementary material.

Similarly, one can prove (21) in Remark 1 for the Markovian case using (24) after adjusting the constants. The upshots of Thm. 2 are: i) it matches the convergence rate of the centralized TD learning in [3] (which implements a less practical projection step though), and [32] (whose bound becomes available only after a mixing-time number of updates); and, ii) it improves upon the existing convergence result $\mathcal{O}(N\alpha)$ of decentralized TD learning (DTDL) reported in [11] (with the projection step) and [33], by removing the scaling factor $N$ from the error term. Thus, to achieve the same accuracy, an about $N$-times smaller stepsize is required by DTDL algorithms than Alg. 1 to reduce the variances present in local updates, which in turn slows down the practical convergence $N$ times too. In contrast, this is not an issue for Alg. 1, which can attain a high-accuracy solution as fast as the centralized TD learning, while respecting data privacy and communication concerns.

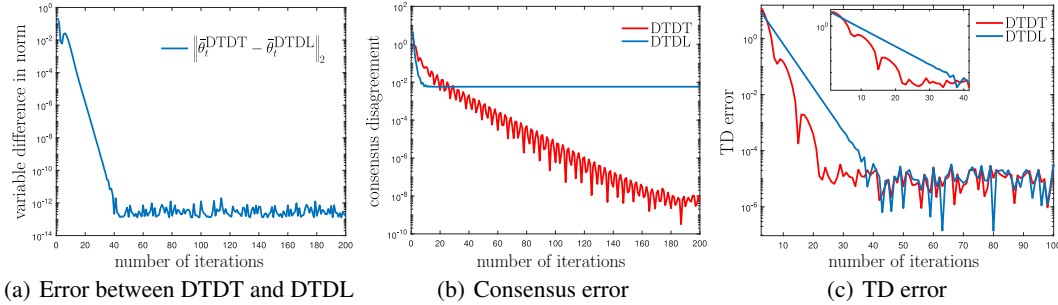

|  |  |  |
|---|---|---|
| (a) Error between DTDT and DTDL | (b) Consensus error | (c) TD error |

Figure 1: Convergence behaviors of the DTDT and DTDL algorithms

## 5 Numerical results

In this section, we present numerical tests to showcase the convergence behaviors of the proposed DTDT as well as the classic DTDL algorithms [26, 11, 33]. The experimental setting is: number of agents $N = 100$, state-space $|\mathcal{S}| = 100$, length of state $|s| = 10$, problem dimension $d = 10$; the communication graph is connected and randomly generated by the Erdős–Rényi model; while the transition matrix $\boldsymbol{P}$ is doubly stochastic, and symmetric. Rewards $\{r(s, a, s') \geq 0\}$ were generated randomly, by following the uniform distribution over $[0, 0.01]$. Feature vectors were generated as $\boldsymbol{\phi}(s) = \cos(\boldsymbol{A}s)$, with entries of $\boldsymbol{A} \in \mathbb{R}^{d \times |s|}$ independently drawn from the Gaussian distribution with zero mean and variance 0.01. Figure 1 shows that, with the same stepsize $\alpha = 0.3$, both the DTDT and DTDL algorithms converge to the same fixed-point solution, but the proposed DTDT achieves a consensus error (disagreement) several orders-of-magnitude smaller than that of DTDT. From the size of TD-error viewpoint, our DTDT is also faster than DTDL. These observations are consistent with our analysis, corroborating that gradient tracking is able to track the average of the network's full gradient efficiently at local nodes.

## 6 Closing discussion

This present paper introduced a decentralized TD tracking (DTDT) technique for multi-agent policy evaluation. Non-asymptotic performance analyses of the proposed DTDT method were provided under both i.i.d. as well as Markov correlated data samples. The novel convergence results match those of the centralized TD learning method, and improve upon those of existing decentralized TD learning algorithms by eliminating their scaling dependence on the number of agents in the limiting error bounds. Although the emphasis here was placed on TD learning for policy evaluation, decentralized variance reduction through gradient tracking can be useful and further explored in more general multi-agent RL settings. In addition, the unifying multistep Lyapunov analysis developed here may also be of independent interest when dealing with learning from Markovian or more generally correlated data.

## Broader impact

In its core, this work contributes to the development and performance analysis of DTDT, a faster multi-agent reinforcement learning (MARL) algorithm for policy evaluation. Given the documented success of MARL in diverse challenging applications such as artificial intelligence [27], quantum computing [28], healthcare [13], and drug design [29], the novel tools will also have major impact in several science and engineering fields, including control, communications and networking, robotics, transportation, neuroscience, as well as medicine and finance. The developed algorithms and tools will thus enable technology transfer to benefit a wide population and improve healthcare and autonomous driving. Taking the pandemic control of COVID-19 as an example, the proposed DTDT technique can be used to provide faster and more accurate outbreak response policies to curb the virus spread in the long term with the least disruption to the economic activity. Although it is capable of boosting public health, the current approach may lead to negative consequences due to privacy disclosure, data leakage, as well as lack of adversarial robustness and fairness guarantees.

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
