[Supplementary Material]

# Supplementary Material for "Decentralized TD Tracking with Linear Function Approximation and its Finite-Time Analysis"

***Remark.*** *The equations* (1)—(24) *and Assumptions 1—4 are referenced with respect to the indexing used in the uploaded paper.*

## A  Proof of Theorem 1

Putting together the bound in (19) and the bound in (17) multiplied by $N^{-1}$, along with the bound in (18) multiplied by $\frac{\alpha|\lambda_1|}{120N(1+1/\tau)(1+\gamma)^2}$, we arrive at the following inequality

$$
\widetilde{\mathcal{W}}_{t+1} := \left(1 - \frac{\alpha|\lambda_1|}{5}\right) \mathbb{E}\left[\left\|\bar{\boldsymbol{\theta}}_{t+1} - \bar{\boldsymbol{\theta}}^*\right\|^2\right] + N^{-1}\mathbb{E}\left[\left\|\boldsymbol{\Theta}_{t+1} - \mathbf{1}\bar{\boldsymbol{\theta}}_{t+1}^\top\right\|^2\right]
$$

$$
+ \frac{\alpha|\lambda_1|}{120(1+1/\tau)(1+\gamma)^2} N^{-1}\mathbb{E}\left[\left\|\boldsymbol{\Psi}_{t+1} - \mathbf{1}\bar{\boldsymbol{\psi}}_{t+1}^\top\right\|^2\right]
$$

$$
\leq \left[1 + 2\alpha\lambda_1 + \alpha|\lambda_1| + 4\alpha^2(1+\gamma)^2\right]\mathbb{E}\left[\left\|\bar{\boldsymbol{\theta}}_t - \bar{\boldsymbol{\theta}}^*\right\|^2\right]
$$

$$
+ \left[(1+\tau)\eta + \frac{\alpha|\lambda_1|}{5} + \frac{4\alpha^2(1+\gamma)^2}{N}\right] N^{-1}\mathbb{E}\left[\left\|\boldsymbol{\Theta}_t - \mathbf{1}\bar{\boldsymbol{\theta}}_t^\top\right\|^2\right]
$$

$$
+ \frac{\alpha|\lambda_1|}{120(1+1/\tau)(1+\gamma)^2}\left[(1+\tau)\eta^2 + \frac{120\alpha(1+1/\tau)^2(1+\gamma)^2}{|\lambda_1|}\right] N^{-1}\mathbb{E}\left[\left\|\boldsymbol{\Psi}_t - \mathbf{1}\bar{\boldsymbol{\psi}}_t^\top\right\|^2\right]
$$

$$
+ \alpha\left[\left(\frac{4|\lambda_1|}{5} + 4\alpha(1+\gamma)^2\right)\|\bar{\boldsymbol{\theta}}^*\|^2 + \frac{|\lambda_1|r_{\max}^2}{20(1+\gamma)^2} + 4\alpha r_{\max}^2\right] \tag{25}
$$

where the stepsize $\alpha > 0$ obeys

$$
1 - \frac{\alpha|\lambda_1|}{5} > 0 \implies 0 < \alpha < \frac{5}{|\lambda_1|}. \tag{26}
$$

Now let us divide both sides of (25) by $1 - \alpha|\lambda_1|/5 > 0$, and we have the following

$$
\mathcal{W}_{t+1} := \mathbb{E}\left[\left\|\bar{\boldsymbol{\theta}}_{t+1} - \bar{\boldsymbol{\theta}}^*\right\|^2\right] + \frac{1}{1-\alpha|\lambda_1|/5} N^{-1}\mathbb{E}\left[\left\|\boldsymbol{\Theta}_{t+1} - \mathbf{1}\bar{\boldsymbol{\theta}}_{t+1}^\top\right\|^2\right]
$$

$$
+ \frac{1}{1-\alpha|\lambda_1|/5} \times \frac{\alpha|\lambda_1|}{120(1+1/\tau)(1+\gamma)^2} N^{-1}\mathbb{E}\left[\left\|\boldsymbol{\Psi}_{t+1} - \mathbf{1}\bar{\boldsymbol{\psi}}_{t+1}^\top\right\|^2\right]
$$

$$
\leq \frac{1 + 2\alpha\lambda_1 + \alpha|\lambda_1| + 4\alpha^2(1+\gamma)^2}{1-\alpha|\lambda_1|/5} \mathbb{E}\left[\left\|\bar{\boldsymbol{\theta}}_t - \bar{\boldsymbol{\theta}}^*\right\|^2\right]
$$

$$
+ \frac{(1+\tau)\eta + \alpha|\lambda_1|/5 + 4\alpha^2(1+\gamma)^2/N}{1-\alpha|\lambda_1|/5} N^{-1}\mathbb{E}\left[\left\|\boldsymbol{\Theta}_t - \mathbf{1}\bar{\boldsymbol{\theta}}_t^\top\right\|^2\right]
$$

$$
+ \frac{120\alpha(1+1/\tau)^2(1+\gamma)^2 + |\lambda_1|(1+\tau)\eta^2}{|\lambda_1|\left(1-\alpha|\lambda_1|/5\right)} \times \frac{\alpha|\lambda_1|}{120(1+1/\tau)(1+\gamma)^2} N^{-1}\mathbb{E}\left[\left\|\boldsymbol{\Psi}_t - \mathbf{1}\bar{\boldsymbol{\psi}}_t^\top\right\|^2\right]
$$

$$
+ \alpha\left[\frac{4|\lambda_1| + 20\alpha(1+\gamma)^2}{5\left(1-\alpha|\lambda_1|/5\right)}\|\bar{\boldsymbol{\theta}}^*\|^2 + \frac{|\lambda_1|r_{\max}^2}{20(1+\gamma)^2\left(1-\alpha|\lambda_1|/5\right)} + \frac{4r_{\max}^2\alpha}{1-\alpha|\lambda_1|/5}\right]. \tag{27}
$$

For notational convenience, define the following coefficients as functions of $\alpha > 0$

$$
\rho_0(\alpha) := \frac{\alpha|\lambda_1|}{120(1+1/\tau)(1+\gamma)^2}, \tag{28a}
$$

$$
\rho_1(\alpha) := \frac{1 + 2\alpha\lambda_1 + \alpha|\lambda_1| + 4\alpha^2(1+\gamma)^2}{1-\alpha|\lambda_1|/5} = \frac{1 - \alpha|\lambda_1| + 4\alpha^2(1+\gamma)^2}{1-\alpha|\lambda_1|/5}, \tag{28b}
$$

$$
\rho_2(\alpha) := \frac{(1+\tau)\eta + \alpha|\lambda_1|/5 + 4\alpha^2(1+\gamma)^2/N}{1-\alpha|\lambda_1|/5}, \tag{28c}
$$

$$\rho_3(\alpha) := \frac{120\alpha(1+1/\tau)^2(1+\gamma)^2 + |\lambda_1|(1+\tau)\eta^2}{|\lambda_1|(1-\alpha|\lambda_1|/5)}, \tag{28d}$$

$$\epsilon_1(\alpha) := \frac{4|\lambda_1| + 20\alpha(1+\gamma)^2}{5(1-\alpha|\lambda_1|/5)}\|\bar{\boldsymbol{\theta}}^*\|^2 + \frac{|\lambda_1|r_{\max}^2}{20(1+\gamma)^2(1-\alpha|\lambda_1|/5)} + \frac{4r_{\max}^2\alpha}{1-\alpha|\lambda_1|/5} \tag{28e}$$

where (28b) uses the fact that $\lambda_1 < 0$. Consider now the following potential function

$$\mathcal{P}_t := \mathbb{E}\big[\|\bar{\boldsymbol{\theta}}_t - \bar{\boldsymbol{\theta}}^*\|^2\big]\,\mathbb{E}\big[\|\boldsymbol{\Theta}_t - \mathbf{1}\bar{\boldsymbol{\theta}}_t^\top\|^2\big] + \rho_0(\alpha)N^{-1}\mathbb{E}\big[\|\boldsymbol{\Psi}_t - \mathbf{1}\bar{\psi}_t^\top\|^2\big] \ge 0. \tag{29}$$

Since the coefficient $\frac{1}{1-\alpha|\lambda_1|/5} > 1$ on the left-hand-side of inequality (27), it can be easily checked from (27) and (29) that

$$\mathcal{P}_{t+1} \le \mathcal{W}_{t+1} \le \max\{\rho_1(\alpha), \rho_2(\alpha), \rho_3(\alpha)\}\,\mathcal{P}_t + \alpha\epsilon_1(\alpha). \tag{30}$$

To guarantee a contracting sequence of $\{P_t\}_t$, besides having $0 < \alpha < \frac{5}{|\lambda_1|}$, it suffices to require the stepsize $\alpha > 0$ such that the following hold

$$\begin{cases} 0 < \rho_1(\alpha) < 1 \\ 0 < \rho_2(\alpha) < 1 \\ 0 < \rho_3(\alpha) < 1 \end{cases} \implies \begin{cases} 0 < \alpha < \frac{|\lambda_1|}{5(1+\gamma)^2} \triangleq \bar{\alpha}_1 \\ 20(1+\gamma)^2\alpha^2 + 2N|\lambda_1|\alpha + 5N[(1+\tau)\eta - 1] < 0 \\ \alpha < \frac{5|\lambda_1|[1-(1+\tau)\eta^2]}{600(1+1/\tau)^2(1+\gamma)^2+|\lambda_1|^2} \end{cases} \tag{31}$$

where $\tau > 0$ can be chosen such that, for any given $0 < \eta < 1$, it holds that i) $(1+\tau)\eta - 1 < 0$, and ii) $1 - (1+\tau)\eta^2 > 0$ in the second, and third inequalities of (31), respectively. For example, if we take $\tau = \frac{1-\eta}{2\eta} > 0$, it is easy to check that $(1+\tau)\eta - 1 = -\frac{1}{2} < 0$, and $1 - (1+\tau)\eta^2 = \frac{2-\eta-\eta^2}{2} > 0$ due again to $0 < \eta < 1$. Then the third inequality reduces to

$$\alpha < \frac{5|\lambda_1|(2-\eta-\eta^2)(1-\eta)^2}{1200(1+\eta)^2(1+\gamma)^2 + 2|\lambda_1|^2(1-\eta)^2} \triangleq \bar{\alpha}_3 \tag{32}$$

In this case, the feasible set for the second inequality is characterized by

$$0 < \alpha < \frac{-N|\lambda_1| + \sqrt{N^2|\lambda_1|^2 + 50N(1+\gamma)^2}}{20(1+\gamma)^2} \triangleq \bar{\alpha}_2. \tag{33}$$

Now combining all these bounds on $\alpha$ gives rise to the following feasible region

$$0 < \alpha < \min\{\bar{\alpha}_1, \bar{\alpha}_2, \bar{\alpha}_3\} \triangleq \bar{\alpha}_i. \tag{34}$$

On the other hand, it can be easily verified that the term $\epsilon_1(\alpha)$ in (28e) is an increasing function of $\alpha > 0$. Thus, it can be upper bounded as follows

$$\epsilon_1(\alpha) \le \epsilon_1(\bar{\alpha}_i) \tag{35}$$

which is now a constant independent of stepsize $\alpha > 0$.

In a nutshell, we conclude that for any stepsize $0 < \alpha < \bar{\alpha}_i$ defined in (34), the potential function $P_t$ satisfies the following contraction

$$\mathcal{P}_{t+1} \le \rho_i(\alpha)\mathcal{P}_t + \alpha\epsilon_1(\bar{\alpha}_i) \tag{36}$$

where $\rho_i(\alpha) := \max\{\rho_1(\alpha), \rho_2(\alpha), \rho_3(\alpha)\} < 1$. It can be deduced that

$$\mathcal{P}_t \le [\rho_i(\alpha)]^t\mathcal{P}_0 + \frac{\alpha\epsilon_1(\bar{\alpha}_i)}{1-\rho_i(\alpha)} \le [\rho_i(\alpha)]^t\mathcal{P}_0 + \frac{\alpha\epsilon_1(\bar{\alpha}_i)}{1-\rho_i(\bar{\alpha}_i)} \triangleq c_0[\rho_i(\alpha)]^t + c_1(\bar{\alpha}_i)\alpha \tag{37}$$

with the constants $c_0 := \mathcal{P}_0 = \|\boldsymbol{\theta}_0 - \bar{\boldsymbol{\theta}}^*\|^2$ and $c_1(\bar{\alpha}_i) := \epsilon_1(\bar{\alpha}_i)/[1-\rho_i(\bar{\alpha}_i)]$ under given initialization in Alg. 1 and assuming that $\boldsymbol{\theta}_0$ is deterministic.

# B  Proof of Theorem 2

Summing the bound in Lemma 2 over a number of $T$ consecutive iterations gives rise to

$$\sum_{k=0}^{T-1}\left\|\boldsymbol{\Theta}_{t+k+1}-\mathbf{1}\bar{\boldsymbol{\theta}}_{t+k+1}^{\top}\right\|^{2}\leq(1+\tau)\eta\sum_{k=0}^{T-1}\left\|\boldsymbol{\Theta}_{t+k}-\mathbf{1}\bar{\boldsymbol{\theta}}_{t+k}^{\top}\right\|^{2}+\alpha^{2}(1+1/\tau)\sum_{k=0}^{T-1}\left\|\boldsymbol{\Psi}_{t+k}-\mathbf{1}\bar{\boldsymbol{\psi}}_{t+k}^{\top}\right\|^{2}$$
(38)

and similarly for the bound in Lemma 6, it follows that

$$\sum_{k=0}^{T-1}\mathbb{E}\left[\left\|\bar{\boldsymbol{\theta}}_{t+k+T}-\boldsymbol{\theta}^{*}\right\|^{2}\right]\leq\left[1+2\alpha T\lambda_{1}+4\alpha T\sigma(T;t)+\alpha^{2}T^{2}(1+\gamma)^{2}\left[3+T\alpha'(1+\gamma)\right]\right.$$

$$+\frac{2\alpha^{4}T^{4}(1+\gamma)^{4}}{9N}[3+T\alpha'(1+\gamma)]^{2}+\frac{8\alpha^{2}T^{2}(1+\gamma)^{2}}{N}\bigg]\sum_{k=0}^{T-1}\mathbb{E}\left[\left\|\bar{\boldsymbol{\theta}}_{t+k}-\boldsymbol{\theta}^{*}\right\|^{2}\right]$$

$$+\left[\frac{\alpha^{2}T^{2}(1+\gamma)^{2}\left[3+T\alpha'(1+\gamma)\right]}{6N}+\frac{2\alpha^{4}T^{4}(1+\gamma)^{4}}{9N^{2}}[3+T\alpha'(1+\gamma)]^{2}\right.$$

$$+\frac{8\alpha^{2}T^{2}(1+\gamma)^{2}}{N^{2}}\bigg]\sum_{k=0}^{T-1}\mathbb{E}\left[\left\|\boldsymbol{\Theta}_{t+k}-\mathbf{1}\bar{\boldsymbol{\theta}}_{t+k}^{\top}\right\|^{2}\right]$$

$$+\left[\frac{\alpha^{2}T^{3}(1+\gamma)^{2}\left[3+T\alpha'(1+\gamma)\right]}{6}+\frac{2\alpha^{4}T^{5}(1+\gamma)^{4}}{9N}[3+T\alpha'(1+\gamma)]^{2}\right.$$

$$+\frac{8\alpha^{2}T^{3}(1+\gamma)^{2}}{N}\bigg]\left\|\bar{\boldsymbol{\theta}}^{*}\right\|^{2}+\frac{\alpha^{2}T^{3}(1+\gamma)^{2}\left[3+T\alpha'(1+\gamma)\right]r_{\max}^{2}}{6N}$$

$$+\frac{2\alpha^{4}T^{5}(1+\gamma)^{4}}{9}[3+T\alpha'(1+\gamma)]^{2}r_{\max}^{2}+8\alpha^{2}T^{3}r_{\max}^{2}+\frac{\alpha T^{2}\sigma(T;t)}{2}$$
(39)

and likewise for Lemma 3, we have that

$$\sum_{k=0}^{T-1}\left\|\boldsymbol{\Psi}_{t+k+1}-\mathbf{1}\bar{\boldsymbol{\psi}}_{t+k+1}^{\top}\right\|^{2}$$

$$\leq(1+\tau)\eta^{2}\sum_{k=0}^{T-1}\left\|\boldsymbol{\Psi}_{t+k}-\mathbf{1}\bar{\boldsymbol{\psi}}_{t+k}^{\top}\right\|^{2}+24(1+1/\tau)(1+\gamma)^{2}\sum_{k=0}^{T-1}\left\|\boldsymbol{\Theta}_{t+k}-\mathbf{1}\bar{\boldsymbol{\theta}}_{t}^{\top}\right\|^{2}$$

$$+24N(1+1/\tau)(1+\gamma)^{2}\sum_{k=0}^{T-1}\left\|\bar{\boldsymbol{\theta}}_{t+k+1}-\bar{\boldsymbol{\theta}}^{*}\right\|^{2}+120N(1+1/\tau)(1+\gamma)^{2}\sum_{k=0}^{T-1}\left\|\bar{\boldsymbol{\theta}}_{t+k}-\bar{\boldsymbol{\theta}}^{*}\right\|^{2}$$

$$+96NT(1+1/\tau)(1+\gamma)^{2}\left\|\bar{\boldsymbol{\theta}}^{*}\right\|^{2}+6NT(1+1/\tau)r_{\max}^{2}$$
(40)

which, upon merging common terms and adding to the right-hand-side $24N(1+1/\tau)(1+\gamma)^{2}\|\bar{\boldsymbol{\theta}}_{t}-\bar{\boldsymbol{\theta}}^{*}\|^{2}$, becomes

$$\sum_{k=0}^{T-1}\left\|\boldsymbol{\Psi}_{t+k+1}-\mathbf{1}\bar{\boldsymbol{\psi}}_{t+k+1}^{\top}\right\|^{2}-24N(1+1/\tau)(1+\gamma)^{2}\left\|\bar{\boldsymbol{\theta}}_{t+T}-\bar{\boldsymbol{\theta}}^{*}\right\|^{2}$$

$$\leq(1+\tau)\eta^{2}\sum_{k=0}^{T-1}\left\|\boldsymbol{\Psi}_{t+k}-\mathbf{1}\bar{\boldsymbol{\psi}}_{t+k}^{\top}\right\|^{2}+24(1+1/\tau)(1+\gamma)^{2}T\left\|\boldsymbol{\Theta}_{t+k}-\mathbf{1}\bar{\boldsymbol{\theta}}_{t}^{\top}\right\|^{2}$$

$$+144N(1+1/\tau)(1+\gamma)^{2}\sum_{k=0}^{T}\left\|\bar{\boldsymbol{\theta}}_{t+k}-\bar{\boldsymbol{\theta}}^{*}\right\|^{2}+96NT(1+1/\tau)(1+\gamma)^{2}\left\|\bar{\boldsymbol{\theta}}^{*}\right\|^{2}+6NT(1+1/\tau)r_{\max}^{2}.$$
(41)

Summing the bound in (39), the bound in (38) multiplied by $1/N$, and the bound in (41) by $\alpha/N$, we arrive at

$$\tilde{\mathcal{Y}}_{t+1} := \sum_{k=0}^{T-1} \mathbb{E}\big[\|\bar{\boldsymbol{\theta}}_{t+k+T} - \bar{\boldsymbol{\theta}}^*\|^2\big] - 24\alpha^2(1+1/\tau)(1+\gamma)^2 \left\|\bar{\boldsymbol{\theta}}_{t+T} - \bar{\boldsymbol{\theta}}^*\right\|^2$$

$$+ N^{-1}\sum_{k=0}^{T-1}\mathbb{E}\big[\|\boldsymbol{\Theta}_{t+k+1} - \mathbf{1}\bar{\boldsymbol{\theta}}_{t+k+1}^\top\|^2\big] + \frac{\alpha}{N}\sum_{k=0}^{T-1}\mathbb{E}\big[\|\boldsymbol{\Psi}_{t+k+1} - \mathbf{1}\bar{\psi}_{t+k+1}^\top\|^2\big]$$

$$\leq \left\{ 1 + 2\alpha T\lambda_1 + 4\alpha T\sigma(T;t) + \alpha^2 T^2(1+\gamma)^2\,[3 + T\alpha'(1+\gamma)] \right.$$

$$+ \frac{2\alpha^4 T^4(1+\gamma)^4}{9N}[3 + T\alpha'(1+\gamma)]^2 + \frac{8\alpha^2 T^2(1+\gamma)^2}{N} \left\} \sum_{k=0}^{T-1}\mathbb{E}\Big[\left\|\bar{\boldsymbol{\theta}}_{t+k} - \bar{\boldsymbol{\theta}}^*\right\|^2\Big] \right.$$

$$+ \left\{ \frac{\alpha^2 T^2(1+\gamma)^2\,[3 + T\alpha'(1+\gamma)]}{6N} + \frac{2\alpha^4 T^4(1+\gamma)^4}{9N^2}[3 + T\alpha'(1+\gamma)]^2 \right.$$

$$+ \frac{8\alpha^2 T^2(1+\gamma)^2}{N^2} \left\} \sum_{k=0}^{T-1}\mathbb{E}\Big[\left\|\boldsymbol{\Theta}_{t+k} - \mathbf{1}\bar{\boldsymbol{\theta}}_{t+k}^\top\right\|^2\Big] \right.$$

$$+ \left\{ \frac{\alpha^2 T^3(1+\gamma)^2\,[3 + T\alpha'(1+\gamma)]}{6} + \frac{2\alpha^4 T^5(1+\gamma)^4}{9N}[3 + T\alpha'(1+\gamma)]^2 \right.$$

$$+ \frac{8\alpha^2 T^3(1+\gamma)^2}{N} \left\} \|\bar{\boldsymbol{\theta}}^*\|^2 + \frac{\alpha^2 T^3(1+\gamma)^2\,[3 + T\alpha'(1+\gamma)]\,r_{\max}^2}{6N} \right.$$

$$+ \frac{2\alpha^4 T^5(1+\gamma)^4}{9}[3 + T\alpha'(1+\gamma)]^2\,r_{\max}^2 + 8\alpha^2 T^3 r_{\max}^2 + \frac{\alpha T^2\sigma(T;t)}{2}$$

$$+ \frac{(1+\tau)\eta}{N}\sum_{k=0}^{T-1}\mathbb{E}\Big[\left\|\boldsymbol{\Theta}_{t+k} - \mathbf{1}\bar{\boldsymbol{\theta}}_{t+k}^\top\right\|^2\Big] + \frac{(1+1/\tau)\alpha^2}{N}\sum_{k=0}^{T-1}\mathbb{E}\Big[\left\|\boldsymbol{\Psi}_{t+k} - \mathbf{1}\bar{\psi}_{t+k}^\top\right\|^2\Big]$$

$$+ \frac{(1+\tau)\alpha\eta^2}{N}\sum_{k=0}^{T-1}\mathbb{E}\Big[\left\|\boldsymbol{\Psi}_{t+k} - \mathbf{1}\bar{\psi}_{t+k}^\top\right\|^2\Big] + \frac{24\alpha T(1+1/\tau)(1+\gamma)^2}{N}\mathbb{E}\big[\|\boldsymbol{\Theta}_t - \mathbf{1}\bar{\boldsymbol{\theta}}_t^\top\|^2\big]$$

$$+ 144\alpha(1+1/\tau)(1+\gamma)^2\sum_{k=0}^{T-1}\mathbb{E}[\|\bar{\boldsymbol{\theta}}_{t+k} - \bar{\boldsymbol{\theta}}^*\|^2] + 96\alpha T(1+1/\tau)(1+\gamma)^2\|\bar{\boldsymbol{\theta}}^*\|^2 + 6\alpha T(1+1/\tau)r_{\max}^2$$

$$\leq \left\{ 1 + 2\alpha T\lambda_1 + 4\alpha T\sigma(T;t) + 144\alpha(1+1/\tau)(1+\gamma)^2 + \alpha^2 T^2(1+\gamma)^2\,[3 + T\alpha'(1+\gamma)] \right.$$

$$+ \frac{2\alpha^4 T^4(1+\gamma)^4}{9N}[3 + T\alpha'(1+\gamma)]^2 + \frac{8\alpha^2 T^2(1+\gamma)^2}{N} \left\} \sum_{k=0}^{T-1}\mathbb{E}\Big[\left\|\bar{\boldsymbol{\theta}}_{t+k} - \bar{\boldsymbol{\theta}}^*\right\|^2\Big] \right.$$

$$+ \frac{1}{18N^2}\left\{ 18N(1+\tau)\eta + 432N\alpha T(1+1/\tau)(1+\gamma)^2 + 3N\alpha^2 T^2(1+\gamma)^2\,[3 + T\alpha'(1+\gamma)] \right.$$

$$+ 2\alpha^4 T^4(1+\gamma)^4[3 + T\alpha'(1+\gamma)]^2 + 144\alpha^2 T^2(1+\gamma)^2 \left\} \sum_{k=0}^{T-1}\mathbb{E}\Big[\left\|\boldsymbol{\Theta}_{t+k} - \mathbf{1}\bar{\boldsymbol{\theta}}_{t+k}^\top\right\|^2\Big] \right.$$

$$+ \frac{\alpha(1+\tau)\eta^2 + \alpha^2(1+1/\tau)}{N}\sum_{k=0}^{T-1}\mathbb{E}\Big[\left\|\boldsymbol{\Psi}_{t+k} - \mathbf{1}\bar{\psi}_{t+k}^\top\right\|^2\Big]$$

$$+ \frac{1}{18N}\left\{ 1728N\alpha T(1+1/\tau)(1+\gamma)^2 + 3N\alpha^2 T^3(1+\gamma)^2[3 + T\alpha'(1+\gamma)] + 144\alpha^2 T^3(1+\gamma)^2 \right.$$

$$+ 4\alpha^4 T^5(1+\gamma)^4[3 + T\alpha'(1+\gamma)]^2 \left\}\|\bar{\boldsymbol{\theta}}^*\|^2 \right.$$

$$+ \frac{\alpha^2 T^3(1+\gamma)^2[3 + T\alpha'(1+\gamma)]r_{\max}^2}{6N} + \frac{2\alpha^4 T^5(1+\gamma)^4}{9}[3 + T\alpha'(1+\gamma)]^2\,r_{\max}^2 + 8\alpha^2 T^3 r_{\max}^2$$

$$+ 6\alpha T(1 + 1/\tau)r_{\max}^2 + \frac{\alpha T^2 \sigma(T;t)}{2}. \tag{42}$$

On the other hand, the left-hand-side of (42) can be lower bounded as follows

$$\tilde{\mathcal{Y}}_{t+1} \geq \left[1 - 24\alpha^2(1 + 1/\tau)(1 + \gamma)^2\right] \sum_{k=0}^{T-1} \mathbb{E}\left[\|\bar{\boldsymbol{\theta}}_{t+k+T} - \bar{\boldsymbol{\theta}}^*\|^2\right]$$

$$+ N^{-1} \sum_{k=0}^{T-1} \mathbb{E}\left[\|\boldsymbol{\Theta}_{t+k+1} - \mathbf{1}\bar{\boldsymbol{\theta}}_{t+k+1}^\top\|^2\right] + \frac{\alpha}{N} \sum_{k=0}^{T-1} \mathbb{E}\left[\|\boldsymbol{\Psi}_{t+k+1} - \mathbf{1}\bar{\boldsymbol{\psi}}_{t+k+1}^\top\|^2\right]. \tag{43}$$

Combining the lower and upper bounds in (42) and (43), followed by dividing both sides by $1 - 24\alpha^2(1 + 1/\tau)(1 + \gamma)^2 > 0$, yields

$$\mathcal{Y}_{t+1} := \sum_{k=0}^{T-1} \mathbb{E}\left[\|\bar{\boldsymbol{\theta}}_{t+k+T} - \bar{\boldsymbol{\theta}}^*\|^2\right] + \frac{1}{1 - 24\alpha^2(1 + 1/\tau)(1 + \gamma)^2} N^{-1} \sum_{k=0}^{T-1} \mathbb{E}\left[\|\boldsymbol{\Theta}_{t+k+1} - \mathbf{1}\bar{\boldsymbol{\theta}}_{t+k+1}^\top\|^2\right]$$

$$+ \frac{1}{1 - 24\alpha^2(1 + 1/\tau)(1 + \gamma)^2} \frac{\alpha}{N} \sum_{k=0}^{T-1} \mathbb{E}\left[\|\boldsymbol{\Psi}_{t+k+1} - \mathbf{1}\bar{\boldsymbol{\psi}}_{t+k+1}^\top\|^2\right]$$

$$\leq \frac{1}{9N[1 - 24\alpha^2(1 + 1/\tau)(1 + \gamma)^2]} \left\{ 9N + 18N\alpha T\lambda_1 + 36N\alpha T\sigma(T;t) + 1296N\alpha(1 + 1/\tau)(1 + \gamma)^2 \right.$$

$$\left. + 72\alpha^2 T^2(1 + \gamma)^2 + 9N\alpha^2 T^2(1 + \gamma)^2[3 + T\alpha'(1 + \gamma)] + 2\alpha^4 T^4(1 + \gamma)^4[3 + T\alpha'(1 + \gamma)]^2 \right\}$$

$$\times \sum_{k=0}^{T-1} \mathbb{E}\left[\|\bar{\boldsymbol{\theta}}_{t+k} - \bar{\boldsymbol{\theta}}^*\|^2\right] + \frac{1}{18N[1 - 24\alpha^2(1 + 1/\tau)(1 + \gamma)^2]} \left\{ 18N(1 + \tau)\eta + 432N\alpha T(1 + 1/\tau)(1 + \gamma)^2 \right.$$

$$\left. + 144\alpha^2 T^2(1 + \gamma)^2 + 3N\alpha^2 T^2(1 + \gamma)^2\left[3 + T\alpha'(1 + \gamma)\right] + 2\alpha^4 T^4(1 + \gamma)^4[3 + T\alpha'(1 + \gamma)]^2 \right\}$$

$$\times N^{-1} \sum_{k=0}^{T-1} \mathbb{E}\left[\|\boldsymbol{\Theta}_{t+k} - \mathbf{1}\bar{\boldsymbol{\theta}}_{t+k}^\top\|^2\right] + \frac{(1 + \tau)\eta^2 + \alpha(1 + 1/\tau)}{1 - 24\alpha^2(1 + 1/\tau)(1 + \gamma)^2} \frac{\alpha}{N} \sum_{k=0}^{T-1} \mathbb{E}\left[\|\boldsymbol{\Psi}_{t+k} - \mathbf{1}\bar{\boldsymbol{\psi}}_{t+k}^\top\|^2\right]$$

$$+ \frac{\alpha}{18N[1 - 24\alpha^2(1 + 1/\tau)(1 + \gamma)^2]} \left\{ 1728NT(1 + 1/\tau)(1 + \gamma)^2 + 3N\alpha T^3(1 + \gamma)^2[3 + T\alpha'(1 + \gamma)] \right.$$

$$\left. + 144\alpha T^3(1 + \gamma)^2 + 4\alpha^3 T^5(1 + \gamma)^4[3 + T\alpha'(1 + \gamma)]^2 \right\} \|\bar{\boldsymbol{\theta}}^*\|^2$$

$$+ \frac{\alpha}{18N[1 - 24\alpha^2(1 + 1/\tau)(1 + \gamma)^2]} \left\{ 108NT(1 + 1/\tau)r_{\max}^2 + 9NT^2\sigma(T;t) + 144N\alpha T^3 r_{\max}^2 \right.$$

$$\left. + 3\alpha T^3(1 + \gamma)^2[3 + T\alpha'(1 + \gamma)]r_{\max}^2 + 4N\alpha^3 T^5(1 + \gamma)^4[3 + T\alpha'(1 + \gamma)]^2 r_{\max}^2 \right\}. \tag{44}$$

For notational brevity, let us define

$$\rho_4(\alpha, T; t) := \frac{1}{9N[1 - 24\alpha^2(1 + 1/\tau)(1 + \gamma)^2]} \left\{ 9N + 18N\alpha T\lambda_1 + 36N\alpha T\sigma(T;t) + 1296N\alpha(1 + 1/\tau)(1 + \gamma)^2 \right.$$

$$\left. + 72\alpha^2 T^2(1 + \gamma)^2 + 9N\alpha^2 T^2(1 + \gamma)^2[3 + T\alpha'(1 + \gamma)] + 2\alpha^4 T^4(1 + \gamma)^4[3 + T\alpha'(1 + \gamma)]^2 \right\}, \tag{45a}$$

$$\rho_5(\alpha, T) := \frac{1}{18N[1 - 24\alpha^2(1 + 1/\tau)(1 + \gamma)^2]} \left\{ 18N(1 + \tau)\eta + 432N\alpha T(1 + 1/\tau)(1 + \gamma)^2 \right.$$

$$\left. + 144\alpha^2 T^2(1 + \gamma)^2 + 3N\alpha^2 T^2(1 + \gamma)^2[3 + T\alpha'(1 + \gamma)] + 2\alpha^4 T^4(1 + \gamma)^4[3 + T\alpha'(1 + \gamma)]^2 \right\}, \tag{45b}$$

$$\rho_6(\alpha, T) := \frac{(1+\tau)\eta^2 + \alpha(1+1/\tau)}{1 - 24\alpha^2(1+1/\tau)(1+\gamma)^2}, \tag{45c}$$

$$\epsilon_2(\alpha, T; t) := \frac{1}{18N[1 - 24\alpha^2(1+1/\tau)(1+\gamma)^2]} \left\{ \left[ 1728N(1+1/\tau)(1+\gamma)^2 + 3N\alpha T^2(1+\gamma)^2[3 + T\alpha'(1+\gamma)] \right. \right.$$
$$\left. + 144\alpha T^2(1+\gamma)^2 + 4\alpha^3 T^4(1+\gamma)^4[3 + T\alpha'(1+\gamma)]^2 \right] \|\bar{\boldsymbol{\theta}}^*\|^2 + 108N(1+1/\tau)r_{\max}^2 + 9NT\sigma(T; t)$$
$$\left. + 144N\alpha T^2 r_{\max}^2 + 3\alpha T^2(1+\gamma)^2[3 + T\alpha'(1+\gamma)]r_{\max}^2 + 4N\alpha^3 T^4(1+\gamma)^4[3 + T\alpha'(1+\gamma)]^2 r_{\max}^2 \right\}. \tag{45d}$$

Consider now the following potential function

$$\mathcal{Q}_t := \sum_{k=0}^{T-1} \mathbb{E}\left[\|\bar{\boldsymbol{\theta}}_{t+k} - \bar{\boldsymbol{\theta}}^*\|^2\right] + N^{-1} \sum_{k=0}^{T-1} \mathbb{E}\left[\|\boldsymbol{\Theta}_{t+k} - \mathbf{1}\bar{\boldsymbol{\theta}}_{t+k}^\top\|^2\right] + \frac{\alpha}{N} \sum_{k=0}^{T-1} \mathbb{E}\left[\|\boldsymbol{\Psi}_{t+k} - \mathbf{1}\bar{\boldsymbol{\psi}}_{t+k}^\top\|^2\right]. \tag{46}$$

Since the coefficient $\frac{1}{1 - 24\alpha^2(1+1/\tau)(1+\gamma)^2} > 1$ for any $0 < \alpha < \frac{1}{\sqrt{24N(1+1/\tau)}(1+\gamma)}$ on the left-hand-side of inequality (44), then one can easily verify that

$$\mathcal{Q}_{t+1} \leq \mathcal{Y}_{t+1} \leq \max\{\rho_4(\alpha, T; t), \rho_5(\alpha, T), \rho_6(\alpha, T)\}\mathcal{Q}_t + \alpha T \epsilon_2(\alpha, T; t). \tag{47}$$

To ensure that the sequence $\{\mathcal{Q}_t\}_t$ is contracting, it suffices to choose the step size $\alpha > 0$ and the parameter $T \in \mathbb{N}^+$ such that the following holds true

$$\max\{\rho_4(\alpha, T; t), \rho_5(\alpha, T), \rho_6(\alpha, T)\} < 1 \implies \begin{cases} 0 < \rho_4(\alpha, T; t) \leq \rho_4(\alpha, T; 0) < 1 \\ 0 < \rho_5(\alpha, T) < 1 \\ 0 < \rho_6(\alpha, T) < 1 \end{cases}$$

yielding

$$1 - 24\alpha^2(1+1/\tau)(1+\gamma)^2 > 0 \tag{48a}$$

$$\underbrace{18N[\lambda_1 T + 2T\sigma(T; 0) + 72(1+1/\tau)(1+\gamma)^2]}_{\triangleq p_3(T)} + \alpha T^2 \underbrace{(1+\gamma)^2[72 + 9N[3 + T\alpha'(1+\gamma)]]}_{\triangleq p_2(T)}$$

$$+ \alpha^3 \underbrace{2T^4(1+\gamma)^4[3 + T\alpha'(1+\gamma)]}_{\triangleq p_1(T)} < 0 \tag{48b}$$

$$\underbrace{18N[(1+\tau)\eta - 1]}_{\triangleq q_4(T)} + \alpha \underbrace{432NT(1+1/\tau)(1+\gamma)^2}_{\triangleq q_3(T)} + \alpha^2 \underbrace{3(1+\gamma)^2[144N(1+1/\tau) + 48T^2 + NT^2[3 + T\alpha'(1+\gamma)]]}_{\triangleq q_2(T)}$$

$$+ \alpha^4 \underbrace{2T^4(1+\gamma)^4[3 + T\alpha'(1+\gamma)]^2}_{\triangleq q_1(T)} < 0 \tag{48c}$$

$$24\alpha^2(1+1/\tau)(1+\gamma)^2 + \alpha(1+1/\tau) + (1+\tau)\eta^2 - 1 < 0. \tag{48d}$$

To ensure that the set of inequalities in (41) has feasible solutions, take $T \in \mathbb{N}^+$ large enough such that

$$\sigma(T; t) \leq \sigma(T; 0) \leq \frac{(1+\gamma)\nu_0}{(1-\rho)T} \max\{2\|\bar{\boldsymbol{\theta}}^*\| + r_{\max}, 1\} \leq \frac{|\lambda_1|}{4}$$

$$\frac{72(1+1/\tau)(1+\gamma)^2}{T} \leq \frac{|\lambda_1|}{4} \tag{49a}$$

namely

$$T \geq \max\left\{ \frac{4\nu_0(1+\gamma)\max\{2\|\bar{\boldsymbol{\theta}}^*\| + r_{\max}, 1\}}{(1-\rho)|\lambda_1|}, \frac{288(1+1/\tau)(1+\gamma)^2}{|\lambda_1|} \right\} \triangleq T_0. \tag{50}$$

For simplicity, we will be working with fixed $T := T_0$ below to derive upper bounds for $\alpha > 0$. Furthermore, take without loss of generality $\tau = \frac{1-\eta}{2\eta} > 0$, and it is easy to check that $(1+\tau)\eta - 1 = -\frac{1}{2} < 0$, and $1 - (1+\tau)\eta^2 = \frac{2-\eta-\eta^2}{2} > 0$ due again to $0 < \eta < 1$. Therefore, the set of inequalities in (48) admits feasible solutions. After tedious algebraic manipulations, a set of necessary conditions on the $\alpha$ for (48) with $T = T_0$ can be given as

$$0 < \alpha < \frac{1}{\sqrt{24(1 + 1/\tau)(1 + \gamma)}} \triangleq \bar{\alpha}_4, \tag{51a}$$

$$0 < \alpha < \sqrt[3]{-\frac{p_3(T_0)}{2p_1(T_0)} + \sqrt{\frac{p_3^2(T_0)}{4p_1^2(T_0)} + \frac{p_2^3(T_0)}{27p_1^3(T_0)}}} + \sqrt[3]{-\frac{p_3(T_0)}{2p_1(T_0)} - \sqrt{\frac{p_3^2(T_0)}{4p_1^2(T_0)} + \frac{p_2^3(T_0)}{27p_1^3(T_0)}}} \triangleq \bar{\alpha}_5(T_0), \tag{51b}$$

$$0 < \alpha < \frac{-\sqrt{2m} + \sqrt{-\frac{2q_2(T_0)}{q_1(T_0)} - 2m + \frac{\sqrt{2}q_3(T_0)}{\sqrt{m}q_1(T_0)}}}{2} \triangleq \bar{\alpha}_6(T_0) \tag{51c}$$

$$0 < \alpha < \frac{-(1 + 1/\tau) + \sqrt{(1 + 1/\tau)^2 + 96(1 + 1/\tau)(1 + \gamma)^2[1 - (1 + \tau)\eta^2]}}{48(1 + 1/\tau)(1 + \gamma)^2} \triangleq \bar{\alpha}_7 \tag{51d}$$

where each and every of the above inequalities constitutes a set of feasible solutions for the corresponding inequality in (48). Moreover, (51b) uses the closed formula of cubic equations and (51c) that of depressed quartic equations with $m$ here being a root of the so-called resolvent cubic of the quartic equation; that is, $8q_1^2(T_0)m^3 + 8q_1(T_0)q_2(T_0)m^2 + [2q_2^2(T_0) - 8q_1(T_0)q_4(T_0)]m - q_3^2(T_0) = 0$ which has closed-form solutions.

When $T = T_0$, a feasible region for the step size $\alpha$ is thus

$$0 < \alpha < \min\{\bar{\alpha}_4, \ \bar{\alpha}_5(T_0), \ \bar{\alpha}_6(T_0), \ \bar{\alpha}_7\} \triangleq \bar{\alpha}_{\mathrm{m}}. \tag{52}$$

On the other hand, one can further check that for fixed $t \in \mathbb{N}$, the function $\epsilon_2(\alpha, T_0; t)$ monotonically increases with $\alpha > 0$; and when fixing $\alpha > 0$, $\epsilon_2(\alpha, T_0; t)$ monotonically decreases with $t > 0$.

In a nutshell, we deduce that for any step size $0 < \alpha < \bar{\alpha}_{\mathrm{m}}$ defined in (52), the potential function $P_t$ satisfies the following contraction

$$\mathcal{Q}_{t+1} \leq \rho_{\mathrm{m}}(\alpha; t)\mathcal{Q}_t + \alpha T_0 \epsilon_2(\alpha, T_0; t) \leq \rho_{\mathrm{m}}(\alpha; 0)\mathcal{Q}_t + \alpha T_0 \epsilon_2(\alpha, T_0; 0) \tag{53}$$

where $\rho_{\mathrm{m}}(\alpha; t) := \max\{\rho_4(\alpha, T_0; t), \ \rho_5(\alpha, T_0), \ \rho_6(\alpha, T_0)\} \leq \rho_{\mathrm{m}}(\alpha; 0) < 1$ for all $t \in \mathbb{N}$. We can then deduce that

$$\mathcal{Q}_t \leq [\rho_{\mathrm{m}}(\alpha; 0)]^t \mathcal{Q}_0 + \frac{T_0 \epsilon_2(\alpha, T_0; 0)}{1 - \rho_{\mathrm{m}}(\alpha; 0)}\alpha \triangleq \mathcal{Q}_0(\alpha, T_0)[\rho_{\mathrm{m}}(\alpha; 0)]^t + c_3(\alpha, T_0)\alpha \tag{54}$$

where $c_3(\alpha, T_0) := T_0 \epsilon_2(\alpha, T_0; 0)/[1 - \rho_{\mathrm{m}}(\alpha; 0)] \leq c_3(\bar{\alpha}_{\mathrm{m}}; T_0)$ since $c_3$ is an increasing function of $\alpha > 0$. Using the definition of $\mathcal{Q}_t$ in (46), we have that

$$\mathcal{Q}_0 = \sum_{k=0}^{T-1} \mathbb{E}\big[\|\bar{\boldsymbol{\theta}}_k - \bar{\boldsymbol{\theta}}^*\|^2\big] + N^{-1} \sum_{k=0}^{T-1} \mathbb{E}\big[\|\boldsymbol{\Theta}_k - \mathbf{1}\bar{\boldsymbol{\theta}}_k^\top\|^2\big] + \frac{\alpha}{N} \sum_{k=0}^{T-1} \mathbb{E}\big[\|\boldsymbol{\Psi}_k - \mathbf{1}\bar{\boldsymbol{\psi}}_k^\top\|^2\big]. \tag{55}$$

Since all stochastic gradients used in the updates (12) from $k = 0$ to $k = T - 1$ as well as the initialization are bounded, then there exists a constant $c_2(\bar{\alpha}_{\mathrm{m}}; T_0)$ such that $\mathcal{Q}_0 \leq c_2(\bar{\alpha}_{\mathrm{m}}; T_0)$. All in all, we conclude that

$$\mathcal{Q}_t \leq c_2(\bar{\alpha}_{\mathrm{m}}; T_0)[\rho_{\mathrm{m}}(\alpha; 0)]^t + c_3(\bar{\alpha}_{\mathrm{m}}, T_0)\alpha \tag{56}$$

which completes the proof.

## C Proof of Lemma 1

Here, we provide a proof of Lemma 1 for completeness, which builds on the matrix Frobenius norm inequality $\|\boldsymbol{AB}\| \leq \sigma_1(\boldsymbol{A})\|\boldsymbol{B}\|$ for any $n \times n$ square matrices $\boldsymbol{A}, \boldsymbol{B}$, with $\sigma_i(\cdot)$ being the $i$-th largest singular value of a matrix. Since $\boldsymbol{W}$ is doubly stochastic, then it follows that

$$\big\|\boldsymbol{W}\boldsymbol{\Theta} - \mathbf{1}\bar{\boldsymbol{\theta}}\big\| = \left\|\Big(\boldsymbol{W} - \frac{1}{N}\mathbf{1}\mathbf{1}^\top\Big)(\boldsymbol{\Theta} - \mathbf{1}\bar{\boldsymbol{\theta}})\right\| \leq \sigma_1\Big(\boldsymbol{W} - \frac{1}{N}\mathbf{1}\mathbf{1}^\top\Big)\big\|\boldsymbol{\Theta} - \mathbf{1}\bar{\boldsymbol{\theta}}\big\|. \tag{57}$$

To prove Lemma 1, it suffices to show that $\sigma_1(\boldsymbol{W} - N^{-1}\mathbf{1}\mathbf{1}^\top) < 1$. To this end, it is clear that

$$\sigma_1^2\left(\boldsymbol{W} - \frac{1}{N}\mathbf{1}\mathbf{1}^\top\right) = \lambda_1\left[\left(\boldsymbol{W} - \frac{1}{N}\mathbf{1}\mathbf{1}^\top\right)\left(\boldsymbol{W} - \frac{1}{N}\mathbf{1}\mathbf{1}^\top\right)^\top\right] = \lambda_1\left(\boldsymbol{W}\boldsymbol{W}^\top - \frac{1}{N}\mathbf{1}\mathbf{1}^\top\right). \quad (58)$$

Recalling that the all-one vector $\mathbf{1}$ is the eigenvector of $\boldsymbol{W}$ associated with its unique largest eigenvalue 1, it follows that $\boldsymbol{W}\boldsymbol{W}^\top\mathbf{1} = \boldsymbol{W}\mathbf{1} = \mathbf{1}$, and based on simple algebraic arguments, we can conclude that $\mathbf{1}$ is the eigenvector of $\boldsymbol{W}\boldsymbol{W}^\top$ associated with its unique largest eigenvalue 1. Thus, we have the eigenvalue decomposition $\boldsymbol{W}\boldsymbol{W}^\top = N^{-1}\mathbf{1}\mathbf{1}^\top + \sum_{i=2}^{N}\lambda_i(\boldsymbol{W}\boldsymbol{W}^\top)\boldsymbol{u}_i\boldsymbol{u}_i^\top$, where $\lambda_2(\boldsymbol{W}\boldsymbol{W}^\top) < 1$ and $\boldsymbol{u}_i$ denotes the normalized eigenvector of $\boldsymbol{W}\boldsymbol{W}^\top$ associated with its $i$-th largest eigenvalue $\lambda_i$. it holds that

$$\lambda_1\left(\boldsymbol{W}\boldsymbol{W}^\top - \frac{1}{N}\mathbf{1}\mathbf{1}^\top\right) = \lambda_2(\boldsymbol{W}\boldsymbol{W}^\top) < 1 \quad (59)$$

which combined with (57) and (58), proves that $\eta = \sigma_1(\boldsymbol{W} - N^{-1}\mathbf{1}\mathbf{1}^\top) = \sqrt{\lambda_2(\boldsymbol{W}\boldsymbol{W}^\top)} < 1$, completing the proof.

## D  Proof of Lemma 2

It is straightforward to check that

$$\left\|\boldsymbol{\Theta}_{t+1} - \mathbf{1}\bar{\boldsymbol{\theta}}_{t+1}^\top\right\|^2 = \left\|\boldsymbol{W}\boldsymbol{\Theta}_t + \alpha\boldsymbol{\Psi}_t - \mathbf{1}\left[\bar{\boldsymbol{\theta}}_t + \alpha\bar{\boldsymbol{\psi}}_t\right]^\top\right\|^2$$

$$\leq (1+\tau)\left\|\boldsymbol{W}\boldsymbol{\Theta}_t - \mathbf{1}\bar{\boldsymbol{\theta}}_t^\top\right\|^2 + (1+1/\tau)\alpha^2\left\|\boldsymbol{\Psi}_t - \mathbf{1}\bar{\boldsymbol{\psi}}_t^\top\right\|^2 \quad (60)$$

$$\leq (1+\tau)\eta\left\|\boldsymbol{\Theta}_t - \mathbf{1}\bar{\boldsymbol{\theta}}_t^\top\right\|^2 + (1+1/\tau)\alpha^2\left\|\boldsymbol{\Psi}_t - \mathbf{1}\bar{\boldsymbol{\psi}}_t^\top\right\|^2 \quad (61)$$

where the first inequality follows from the fact that $2\mathrm{Tr}(\boldsymbol{X}^\top\boldsymbol{Y}) \leq \beta\|\boldsymbol{X}\|^2 + (1/\beta)\|\boldsymbol{Y}\|^2$ for any $\beta \geq 0$; and the second uses Lemma 1.

## E  Proof of Lemma 3

Clearly, it follows that

$$\left\|\boldsymbol{\Psi}_{t+1} - \mathbf{1}\bar{\boldsymbol{\psi}}_{t+1}^\top\right\|^2 = \left\|\boldsymbol{W}\boldsymbol{\Psi}_t + \boldsymbol{G}(\boldsymbol{\Theta}_{t+1}, \zeta_{t+1}) - \boldsymbol{G}(\boldsymbol{\Theta}_t, \zeta_t)\right.$$

$$\left. - \frac{\mathbf{1}\mathbf{1}^\top}{N}\left[\boldsymbol{W}\boldsymbol{\Psi}_t + \boldsymbol{G}(\boldsymbol{\Theta}_{t+1}, \zeta_{t+1}) - \boldsymbol{G}(\boldsymbol{\Theta}_t, \zeta_t)\right]\right\|^2$$

$$= \left\|\left(\boldsymbol{W} - \frac{\mathbf{1}\mathbf{1}^\top}{N}\right)\boldsymbol{\Psi}_t + \left(\boldsymbol{I} - \frac{\mathbf{1}\mathbf{1}^\top}{N}\right)[\boldsymbol{G}(\boldsymbol{\Theta}_{t+1}, \zeta_{t+1}) - \boldsymbol{G}(\boldsymbol{\Theta}_t, \zeta_t)]\right\|^2 \quad (62)$$

$$\leq (1+\tau)\left\|\left(\boldsymbol{W} - \frac{\mathbf{1}\mathbf{1}^\top}{N}\right)\left[\boldsymbol{\Psi}_t - \mathbf{1}\bar{\boldsymbol{\psi}}_t^\top\right]\right\|^2$$

$$+ (1+1/\tau)\left\|\left(\boldsymbol{I} - \frac{\mathbf{1}\mathbf{1}^\top}{N}\right)[\boldsymbol{G}(\boldsymbol{\Theta}_{t+1}, \zeta_{t+1}) - \boldsymbol{G}(\boldsymbol{\Theta}_t, \zeta_t)]\right\|^2 \quad (63)$$

$$\leq (1+\tau)\eta^2\left\|\boldsymbol{\Psi}_t - \mathbf{1}\bar{\boldsymbol{\psi}}_t^\top\right\|^2 + (1+1/\tau)\left\|\boldsymbol{\Theta}_{t+1}\boldsymbol{A}^\top(\zeta_{t+1})\right.$$

$$\left. - \boldsymbol{\Theta}_t\boldsymbol{A}^\top(\zeta_t) + \boldsymbol{r}_{t+1}\boldsymbol{\phi}^\top(s_{t+1}) - \boldsymbol{r}_t\boldsymbol{\phi}^\top(s_t)\right\|^2 \quad (64)$$

$$\leq (1+\tau)\eta^2\left\|\boldsymbol{\Psi}_t - \mathbf{1}\bar{\boldsymbol{\psi}}_t^\top\right\|^2 + 6Nr_{\max}^2(1+1/\tau)$$

$$+ 3(1+1/\tau)\left\|\boldsymbol{\Theta}_{t+1}\boldsymbol{A}^\top(\zeta_{t+1}) - \boldsymbol{\Theta}_t\boldsymbol{A}^\top(\zeta_t)\right\|^2 \quad (65)$$

where (62) merges terms and uses the fact that $\boldsymbol{W}\mathbf{1} = \mathbf{1}$; (63) follows from Young's inequality and the fact that $\left(\boldsymbol{W} - \mathbf{1}\mathbf{1}^\top/N\right)\mathbf{1}\bar{\boldsymbol{\psi}}^\top = \mathbf{0}$; (64) uses the definition of $\boldsymbol{G}(\boldsymbol{\Theta}_t)$; and the last inequality is due to $|r^n| \leq r_{\max}$ and $\|\boldsymbol{\phi}(s)\| \leq 1$ for all $n \in \mathcal{N}$ and $s \in \mathcal{S}$.

For the second term in (65), we have that

$$\left\|\boldsymbol{\Theta}_{t+1}\boldsymbol{A}^\top(\zeta_{t+1}) - \boldsymbol{\Theta}_t\boldsymbol{A}^\top(\zeta_t)\right\|^2 = \left\|[\boldsymbol{W}\boldsymbol{\Theta}_t + \alpha\boldsymbol{\Psi}_t]\boldsymbol{A}^\top(\zeta_{t+1}) - \boldsymbol{\Theta}_t\boldsymbol{A}^\top(\zeta_t)\right\|^2$$

$$= \left\| \boldsymbol{W}\left[\boldsymbol{\Theta}_t - \mathbf{1}\bar{\boldsymbol{\theta}}_t^\top\right] \boldsymbol{A}^\top(\zeta_{t+1}) - \left[\boldsymbol{\Theta}_t - \mathbf{1}\bar{\boldsymbol{\theta}}_t^\top\right] \boldsymbol{A}^\top(\zeta_t) \right.$$
$$\left. + \alpha\boldsymbol{\Psi}_t \boldsymbol{A}^\top(\zeta_{t+1}) + \mathbf{1}\bar{\boldsymbol{\theta}}_t^\top\left[\boldsymbol{A}(\zeta_{t+1}) - \boldsymbol{A}(\zeta_t)\right]^\top \right\|^2$$

$$\leq 4\left\| \boldsymbol{W}\left[\boldsymbol{\Theta}_t - \mathbf{1}\bar{\boldsymbol{\theta}}_t^\top\right] \boldsymbol{A}^\top(\zeta_{t+1})\right\|^2 + 4\left\|\left[\boldsymbol{\Theta}_t - \mathbf{1}\bar{\boldsymbol{\theta}}_t^\top\right] \boldsymbol{A}^\top(\zeta_t)\right\|^2$$
$$+ 4\alpha^2\left\|\boldsymbol{\Psi}_t \boldsymbol{A}^\top(\zeta_{t+1})\right\|^2 + 4\left\|\mathbf{1}\bar{\boldsymbol{\theta}}_t^\top\left[\boldsymbol{A}(\zeta_{t+1}) - \boldsymbol{A}(\zeta_t)\right]^\top\right\|^2 \tag{66}$$

$$\leq 8(1+\gamma)^2\left\|\boldsymbol{\Theta}_t - \mathbf{1}\bar{\boldsymbol{\theta}}_t^\top\right\|^2 + 4\alpha^2(1+\gamma)^2 N\left\|\bar{\boldsymbol{\psi}}_t\right\|^2$$
$$+ 16(1+\gamma)^2 N\left\|\bar{\boldsymbol{\theta}}_t - \bar{\boldsymbol{\theta}}^* + \bar{\boldsymbol{\theta}}^*\right\|^2 \tag{67}$$

$$\leq 8(1+\gamma)^2\left\|\boldsymbol{\Theta}_t - \mathbf{1}\bar{\boldsymbol{\theta}}_t^\top\right\|^2 + 8N(1+\gamma)^2\left[\left\|\bar{\boldsymbol{\theta}}_{t+1} - \bar{\boldsymbol{\theta}}^*\right\|^2 + \left\|\bar{\boldsymbol{\theta}}_t - \bar{\boldsymbol{\theta}}^*\right\|^2\right]$$
$$+ 32N(1+\gamma)^2\left\|\bar{\boldsymbol{\theta}}_t - \bar{\boldsymbol{\theta}}^*\right\|^2 + 32N(1+\gamma)^2\left\|\bar{\boldsymbol{\theta}}^*\right\|^2 \tag{68}$$

$$= 8(1+\gamma)^2\left\|\boldsymbol{\Theta}_t - \mathbf{1}\bar{\boldsymbol{\theta}}_t^\top\right\|^2 + 8N(1+\gamma)^2\left\|\bar{\boldsymbol{\theta}}_{t+1} - \bar{\boldsymbol{\theta}}^*\right\|^2$$
$$+ 40N(1+\gamma)^2\left\|\bar{\boldsymbol{\theta}}_t - \bar{\boldsymbol{\theta}}^*\right\|^2 + 32N(1+\gamma)^2\left\|\bar{\boldsymbol{\theta}}^*\right\|^2 \tag{69}$$

where (66) calls for the result $\|\sum_{n=1}^N \boldsymbol{a}_i\|^2 \leq N\sum_{n=1}^N\|\boldsymbol{a}_i\|^2$; (67) follows from the fact that the spectral radius $\tau(\boldsymbol{A}(\zeta_t)) \leq 1 + \gamma$ and $\|\boldsymbol{\Psi}_t\|^2 \leq N\|\bar{\boldsymbol{\psi}}_t\|^2$ for all $t \in \mathbb{N}^+$; and (68) uses Young's inequality and the following result

$$\left\|\alpha\bar{\boldsymbol{\psi}}_t\right\|^2 = \left\|\bar{\boldsymbol{\theta}}_{t+1} - \bar{\boldsymbol{\theta}}_t\right\|^2$$
$$= \left\|\bar{\boldsymbol{\theta}}_{t+1} - \bar{\boldsymbol{\theta}}^* + \bar{\boldsymbol{\theta}}^* - \bar{\boldsymbol{\theta}}_t\right\|^2$$
$$\leq 2\left\|\bar{\boldsymbol{\theta}}_{t+1} - \bar{\boldsymbol{\theta}}^*\right\|^2 + 2\left\|\bar{\boldsymbol{\theta}}_t - \bar{\boldsymbol{\theta}}^*\right\|^2. \tag{70}$$

Substituting the bound in (69) into (65) and merging common terms, concludes the proof of Lemma 3.

## F   Proof of Lemma 4

Recalling the definitions $\boldsymbol{g}(\bar{\boldsymbol{\theta}}^*) = \boldsymbol{A}\bar{\boldsymbol{\theta}}^* + \bar{\boldsymbol{b}} = \mathbf{0}$ and $\bar{\boldsymbol{\psi}}_t = \bar{\boldsymbol{g}}(\bar{\boldsymbol{\theta}}_t, \zeta_t)$, we have the following result

$$\bar{\boldsymbol{\psi}}_t = N^{-1}\boldsymbol{\Psi}_t^\top\mathbf{1} = N^{-1}\boldsymbol{G}^\top(\boldsymbol{\Theta}_t, \zeta_t)\mathbf{1} \tag{71}$$
$$= N^{-1}\left[\boldsymbol{A}(\zeta_t)\boldsymbol{\Theta}_t^\top + \boldsymbol{\phi}(s_t)\boldsymbol{r}_t^\top\right]\mathbf{1} \tag{72}$$
$$= \boldsymbol{A}(\zeta_t)\bar{\boldsymbol{\theta}}_t + \boldsymbol{\phi}(s_t)\bar{r}_t \triangleq \bar{\boldsymbol{g}}(\bar{\boldsymbol{\theta}}_t, \zeta_t) \tag{73}$$

where (71) holds true when $\boldsymbol{\Psi}(0) = \mathbf{0}$; and $\bar{r}_t := \sum_{n=1}^N r_t^n/N$. Furthermore, it holds that

$$\mathbb{E}_\pi[\boldsymbol{\phi}(s_t)\bar{r}_t|\mathcal{F}_t] = N^{-1}\sum_{n=1}^N \mathbb{E}_\pi[r_t^n\boldsymbol{\phi}(s_t)|\mathcal{F}_t] = N^{-1}\sum_{n=1}^N \boldsymbol{b}^n = \bar{\boldsymbol{b}}. \tag{74}$$

**Lemma 7.** *The average $(1/N)\boldsymbol{G}^\top(\boldsymbol{\Theta}_t, \zeta_t)\mathbf{1}$ of the gradient estimates at all agents is an unbiased estimate of $\boldsymbol{g}(\bar{\boldsymbol{\theta}}_t)$ for any given $\boldsymbol{\Theta}_t \in \mathbb{R}^{N \times d}$; that is,*

$$\mathbb{E}_\pi\left[N^{-1}\boldsymbol{G}^\top(\boldsymbol{\Theta}_t, \zeta_t)\mathbf{1} - \boldsymbol{g}(\bar{\boldsymbol{\theta}}_t)\Big|\boldsymbol{\Theta}_t\right] = \mathbf{0} \tag{75}$$

*and the variance satisfies*

$$\mathbb{E}_\pi\left[\left\|N^{-1}\boldsymbol{G}^\top(\boldsymbol{\Theta}_t, \zeta_t)\mathbf{1} - \boldsymbol{g}(\bar{\boldsymbol{\theta}}_t)\right\|^2\Big|\boldsymbol{\Theta}_t\right] \leq 4\beta^2\|\bar{\boldsymbol{\theta}}_t - \bar{\boldsymbol{\theta}}^*\|^2 + 4\beta^2\|\bar{\boldsymbol{\theta}}^*\|^2 + 8r_{\max}^2 \tag{76}$$

*where $\beta$ is the maximum spectral radius of matrices $\{\boldsymbol{A}(\zeta_t) - \boldsymbol{A}\}_{t \in \mathcal{N}^+}$.*

On the other hand, it clearly holds that

$$\mathbb{E}_\pi\left[\left\|\bar{\boldsymbol{\theta}}_{t+1} - \bar{\boldsymbol{\theta}}^*\right\|^2\Big|\boldsymbol{\Theta}_t\right] = \mathbb{E}_\pi\left[\left\|\bar{\boldsymbol{\theta}}_t + \alpha\bar{\boldsymbol{\psi}}_t - \bar{\boldsymbol{\theta}}^*\right\|^2\Big|\boldsymbol{\Theta}_t\right]$$

$$= \left\| \bar{\theta}_t - \bar{\theta}^* \right\|^2 + 2\alpha \langle \bar{\theta}_t - \bar{\theta}^*, \mathbb{E}_\pi \left[ \bar{g}(\bar{\theta}_t, \zeta_t) - g(\bar{\theta}^*) | \Theta_t \right] \rangle$$
$$+ \alpha^2 \mathbb{E}_\pi \left[ \left\| \bar{\psi}_t - A(\zeta_t)\bar{\theta}_t + A(\zeta_t)\bar{\theta}_t \right\|^2 \Big| \Theta_t \right] \tag{77}$$

where (77) appeals to the fact that $g(\bar{\theta}^*) = 0$.

In the following, we will bound the last two terms in (77).

*Bounding the inner product term in* (77). Clearly, it follows that
$$\langle \bar{\theta}_t - \bar{\theta}^*, \mathbb{E}_\pi \left[ \bar{g}(\bar{\theta}_t, \zeta_t) - g(\bar{\theta}^*) | \Theta_t \right] \rangle = \langle \bar{\theta}_t - \bar{\theta}^*, \mathbb{E}_\pi \left[ A(\zeta_t)\bar{\theta}_t + \phi(s_t)\bar{r}_t - A\bar{\theta}^* - \bar{b} | \Theta_t \right] \rangle$$
$$= \langle \bar{\theta}_t - \bar{\theta}^*, A \left[ \bar{\theta}_t - \bar{\theta}^* \right] \rangle \tag{78}$$
$$\leq \lambda_1 \left\| \bar{\theta}_t - \bar{\theta}^* \right\|^2 \tag{79}$$

which (78) uses Lemma 7; and (79) follows from the fact that $A \preceq 0$ with eigenvalues $\lambda_p(A) \leq \cdots \leq \lambda_1 < 0$.

*Bounding the last term in* (77). Using the definition of $\bar{\psi}_t$ in (72), it can be easily verified that

$$\mathbb{E}_\pi \left[ \left\| \bar{\psi}_t - A(\zeta_t)\bar{\theta}_t \right\|^2 \Big| \Theta_t \right] = \mathbb{E}_\pi \left[ \left\| N^{-1} \sum_{n=1}^N \left[ A(\zeta_t)\theta_t^n + \phi(s_t)r_t^n \right] - A(\zeta_t)\bar{\theta}_t \right\|^2 \Big| \Theta_t \right]$$

$$= \mathbb{E}_\pi \left[ \left\| N^{-1} \sum_{n=1}^N \left[ A(\zeta_t)\theta_t^n - A(\zeta_t)\bar{\theta}_t \right] + \phi(s_t)\bar{r}_t \right\|^2 \Big| \Theta_t \right]$$

$$\leq \frac{2}{N^2} \mathbb{E}_\pi \left[ \left\| A(\zeta_t) \sum_{n=1}^N \left( \theta_t^n - \bar{\theta}_t \right) \right\|^2 \Big| \Theta_t \right] + 2 \left\| \phi(s_t)\bar{r}_t \right\|^2 \tag{80}$$

$$\leq \frac{2(1+\gamma)^2}{N^2} \left\| \Theta_t - \mathbf{1}\bar{\theta}_t^\top \right\|^2 + 2r_{\max}^2 \tag{81}$$

where (80) applies Young's inequality; and (81) is due to the facts that $\tau(A(\zeta_t)) \leq 1 + \gamma$, $\bar{r}_t = \sum_{n=1}^N r_t^n / N \leq 1$, and $\|\phi(s_t)\| \leq 1$ for all $s \in \mathcal{S}$.

Substituting the upper bounds in (79) and (81) into (77) as well as using $\alpha^2 \|A(\zeta_t)\bar{\theta}_t\|^2 \leq \alpha^2(1+\gamma)^2 [2\|\bar{\theta}_t - \bar{\theta}^*\|^2 + 2\|\bar{\theta}^*\|^2]$, yields

$$\mathbb{E}_\pi \left[ \left\| \bar{\theta}_{t+1} - \bar{\theta}^* \right\|^2 \Big| \Theta_t \right] \leq \left[ 1 + 2\alpha\lambda_1 + 4\alpha^2(1+\gamma)^2 \right] \left\| \bar{\theta}_t - \bar{\theta}^* \right\|^2 + \frac{4\alpha^2(1+\gamma)^2}{N^2} \left\| \Theta_t - \mathbf{1}\bar{\theta}_t^\top \right\|^2$$
$$+ 4\alpha^2(1+\gamma)^2 \|\bar{\theta}^*\|^2 + 4\alpha^2 r_{\max}^2. \tag{82}$$

Taking expectation of both sides in (82) over $\mathcal{F}_t$ asserts that

$$\mathbb{E} \left[ \left\| \bar{\theta}_{t+1} - \bar{\theta}^* \right\|^2 \right] \leq \left[ 1 + 2\alpha\lambda_1 + 4\alpha^2(1+\gamma)^2 \right] \mathbb{E} \left[ \left\| \bar{\theta}_t - \bar{\theta}^* \right\|^2 \right] + \frac{4\alpha^2(1+\gamma)^2}{N^2} \mathbb{E} \left[ \left\| \Theta_t - \mathbf{1}\bar{\theta}_t^\top \right\|^2 \right]$$
$$+ 4\alpha^2(1+\gamma)^2 \|\bar{\theta}^*\|^2 + 4\alpha^2 r_{\max}^2 \tag{83}$$

concluding the proof.

# G  Proof of Lemma 5

The proof builds upon that of [33, Lem. 2]. For notational brevity, let $r_{\mathcal{G}}(t) := (1/N) \sum_{n \in \mathcal{N}_n} r_t^n$ for each $t \geq 0$. It then follows that

$$\left\| \frac{1}{TN} \sum_{j=t}^{t+T-1} \mathbb{E} \left[ G^\top(\Theta, \zeta_j)\mathbf{1} | \zeta_0 \right] - \bar{g}(\bar{\theta}) \right\|$$

$$= \left\| \frac{1}{T} \sum_{j=t}^{t+T-1} \mathbb{E} \left[ \phi(s_t)[\gamma\phi(s_{t+1}) - \phi(s_t)]^\top \bar{\theta} + \frac{1}{N}\phi(s_t)r_t^\top \mathbf{1} \Big| \zeta_0 \right] - \mathbb{E}_\pi \left[ g(\bar{\theta}) \right] \right\|$$

$$= \left\| \frac{1}{T} \sum_{j=t}^{t+T-1} \sum_{s \in \mathcal{S}} \left( \Pr[s_j = s | s_1 = s'] - \pi(s) \right) \left[ \phi(s) \left( \gamma p(s, s') \phi(s') - \phi(s) \right)^\top (\bar{\boldsymbol{\theta}} + \boldsymbol{\theta}^*) + r_{\mathcal{G}}(s) \phi(s) \right] \right\|$$

$$\leq \max_{s,s'} \left\| \phi(s) \left[ \gamma p(s, s') \phi(s') - \phi(s) \right]^\top (\bar{\boldsymbol{\theta}} + \boldsymbol{\theta}^*) + r_{\mathcal{G}}(s) \phi(s) \right\|$$

$$\times \frac{1}{T} \sum_{j=t}^{t+T-1} \sum_{s \in \mathcal{S}} \left| \Pr[s_j = s | s_1 = s'] - \pi(s) \right|$$

$$\leq (1 + \gamma) \left( \| \bar{\boldsymbol{\theta}} - \boldsymbol{\theta}^* \| + 2 \| \boldsymbol{\theta}^* \| + r_{\max} \right) \times \frac{1}{T} \sum_{j=t}^{t+T-1} \nu_0 \rho^j$$

$$\leq \frac{(1 + \gamma) \nu_0 \rho^t}{(1 - \rho) T} \left( \| \bar{\boldsymbol{\theta}} - \boldsymbol{\theta}^* \| + 2 \| \boldsymbol{\theta}^* \| + r_{\max} \right)$$

$$\leq \sigma(T; t) \left( \| \bar{\boldsymbol{\theta}} - \boldsymbol{\theta}^* \| + 1 \right) \tag{84}$$

where we have defined $\sigma(T; t) = \frac{(1+\gamma)\nu_0 \rho^t}{(1-\rho)T} \times \max\{2\|\boldsymbol{\theta}^*\| + r_{\max}, 1\}$, and the second inequality follows from the fact that any finite-state, irreducible, and aperiodic Markov chains converges geometrically fast (with some initial constant $\nu_0 > 0$ and rate $0 < \rho < 1$) to its unique stationary distribution [21, Thm. 4.9]. Hence, we conclude that Lemma 5 holds with function $\sigma(T)$ monotonically decreasing in $T \in \mathbb{N}^+$.

## H  Proof of Lemma 6

Consider now the following multi-step Lyapunov function

$$\mathcal{V}_t = \sum_{k=t}^{t+T-1} \left\| \bar{\boldsymbol{\theta}}_t - \bar{\boldsymbol{\theta}}^* \right\|^2 \tag{85}$$

from which we have

$$\mathcal{V}_{t+1} - \mathcal{V}_t = \left\| \bar{\boldsymbol{\theta}}_{t+T} - \bar{\boldsymbol{\theta}}^* \right\|^2 - \left\| \bar{\boldsymbol{\theta}}_t - \bar{\boldsymbol{\theta}}^* \right\|^2. \tag{86}$$

According to (12), it is clear that

$$\bar{\boldsymbol{\theta}}_{t+T} = \bar{\boldsymbol{\theta}}_t + \alpha \sum_{k=t}^{t+T-1} \bar{\boldsymbol{\psi}}_k = \bar{\boldsymbol{\theta}}_t + \alpha \sum_{k=t}^{t+T-1} \bar{\boldsymbol{g}}(\bar{\boldsymbol{\theta}}_k, \zeta_k) \tag{87}$$

Therefore, it follows that

$$\left\| \bar{\boldsymbol{\theta}}_{t+T} - \bar{\boldsymbol{\theta}}^* \right\|^2 = \left\| \bar{\boldsymbol{\theta}}_t + \alpha \sum_{k=t}^{t+T-1} \bar{\boldsymbol{\psi}}_k - \bar{\boldsymbol{\theta}}^* \right\|^2$$

$$= \left\| \bar{\boldsymbol{\theta}}_t - \bar{\boldsymbol{\theta}}^* \right\|^2 + 2\alpha \left\langle \bar{\boldsymbol{\theta}}_t - \bar{\boldsymbol{\theta}}^*, \sum_{k=t}^{t+T-1} \bar{\boldsymbol{g}}(\bar{\boldsymbol{\theta}}_k, \zeta_k) \right\rangle + \alpha^2 \left\| \sum_{k=t}^{t+T-1} \bar{\boldsymbol{g}}(\bar{\boldsymbol{\theta}}_k, \zeta_k) \right\|^2$$

$$= \left\| \bar{\boldsymbol{\theta}}_t - \bar{\boldsymbol{\theta}}^* \right\|^2 + 2\alpha \left\langle \bar{\boldsymbol{\theta}}_t - \bar{\boldsymbol{\theta}}^*, T\boldsymbol{g}(\bar{\boldsymbol{\theta}}_t) + \sum_{k=t}^{t+T-1} \left[ \bar{\boldsymbol{g}}(\bar{\boldsymbol{\theta}}_k, \zeta_k) - \bar{\boldsymbol{g}}(\bar{\boldsymbol{\theta}}_t, \zeta_k) \right. \right.$$

$$\left. \left. + \bar{\boldsymbol{g}}(\bar{\boldsymbol{\theta}}_t, \zeta_k) \right] - T\boldsymbol{g}(\bar{\boldsymbol{\theta}}_t) \right\rangle + \alpha^2 \left\| \sum_{k=t}^{t+T-1} \bar{\boldsymbol{g}}(\bar{\boldsymbol{\theta}}_k, \zeta_k) - \bar{\boldsymbol{g}}(\bar{\boldsymbol{\theta}}_t, \zeta_k) + \bar{\boldsymbol{g}}(\bar{\boldsymbol{\theta}}_t, \zeta_k) \right\|^2$$

$$= \left\| \bar{\boldsymbol{\theta}}_t - \bar{\boldsymbol{\theta}}^* \right\|^2 + 2\alpha \left\langle \bar{\boldsymbol{\theta}}_t - \bar{\boldsymbol{\theta}}^*, T\boldsymbol{g}(\bar{\boldsymbol{\theta}}_t) - T\boldsymbol{g}(\bar{\boldsymbol{\theta}}^*) \right\rangle \tag{88a}$$

$$+ 2\alpha \left\langle \bar{\boldsymbol{\theta}}_t - \bar{\boldsymbol{\theta}}^*, \sum_{k=t}^{t+T-1} \left[ \bar{\boldsymbol{g}}(\bar{\boldsymbol{\theta}}_k, \zeta_k) - \bar{\boldsymbol{g}}(\bar{\boldsymbol{\theta}}_t, \zeta_k) \right] \right\rangle \tag{88b}$$

$$+ 2\alpha \left\langle \bar{\boldsymbol{\theta}}_t - \bar{\boldsymbol{\theta}}^*, \sum_{k=t}^{t+T-1} \bar{\boldsymbol{g}}(\bar{\boldsymbol{\theta}}_t, \zeta_k) - T\boldsymbol{g}(\bar{\boldsymbol{\theta}}_t) \right\rangle \tag{88c}$$

$$+ \alpha^2 \left\| \sum_{k=t}^{t+T-1} \left[ \bar{\boldsymbol{g}}(\bar{\boldsymbol{\theta}}_k, \zeta_k) - \bar{\boldsymbol{g}}(\bar{\boldsymbol{\theta}}_t, \zeta_k) + \bar{\boldsymbol{g}}(\bar{\boldsymbol{\theta}}_t, \zeta_k) \right] \right\|^2 \tag{88d}$$

where the third equality follows from adding and subtracting the same terms, and the last from the fact that $\boldsymbol{g}(\bar{\boldsymbol{\theta}}^*) = \boldsymbol{0}$.

In the sequel, we will upper bound each of the terms in (88a)–(88d).

*Bounding the second term in* (88a).

$$\left\langle \bar{\boldsymbol{\theta}}_t - \bar{\boldsymbol{\theta}}^*, T\boldsymbol{g}(\bar{\boldsymbol{\theta}}_t) - T\boldsymbol{g}(\bar{\boldsymbol{\theta}}^*) \right\rangle = T\left\langle \bar{\boldsymbol{\theta}}_t - \bar{\boldsymbol{\theta}}^*, \boldsymbol{A}\bar{\boldsymbol{\theta}}_t + \bar{\boldsymbol{b}} - \boldsymbol{A}\bar{\boldsymbol{\theta}}^* - \bar{\boldsymbol{b}} \right\rangle$$
$$= T\left\langle \bar{\boldsymbol{\theta}}_t - \bar{\boldsymbol{\theta}}^*, \boldsymbol{A}(\bar{\boldsymbol{\theta}}_t - \bar{\boldsymbol{\theta}}^*) \right\rangle$$
$$\leq T\lambda_1 \left\| \bar{\boldsymbol{\theta}}_t - \bar{\boldsymbol{\theta}}^* \right\|^2 \tag{89}$$

where the last inequality is due to $\boldsymbol{A} \preceq \boldsymbol{0}$.

*Bounding the term in* (88b). Recall first that

$$\bar{\boldsymbol{g}}(\bar{\boldsymbol{\theta}}_t, \zeta_t) = N^{-1}\boldsymbol{G}^\top(\boldsymbol{\Theta}_t, \zeta_t)\boldsymbol{1} = N^{-1}\left[ \boldsymbol{A}(\zeta_t)\boldsymbol{\Theta}_t^\top + \boldsymbol{\phi}(s_t)\boldsymbol{r}_t^\top \right]\boldsymbol{1}. \tag{90}$$

Let us define the following function

$$\mathcal{Y}(\bar{\boldsymbol{\theta}}_t, T) := \left\| \sum_{k=t}^{t+T-1} \left[ \bar{\boldsymbol{g}}(\bar{\boldsymbol{\theta}}_k, \zeta_k) - \bar{\boldsymbol{g}}(\bar{\boldsymbol{\theta}}_t, \zeta_k) \right] \right\|$$
$$= N^{-1} \left\| \sum_{k=t}^{t+T-1} \left[ \boldsymbol{A}(\zeta_k)\boldsymbol{\Theta}_k^\top \boldsymbol{1} - \boldsymbol{A}(\zeta_k)\boldsymbol{\Theta}_t^\top \boldsymbol{1} \right] \right\|. \tag{91}$$

Then it readily follows that

$$\mathcal{Y}(\bar{\boldsymbol{\theta}}_t, T) = \left\| \bar{\boldsymbol{g}}(\bar{\boldsymbol{\theta}}_{t+T-1}, \zeta_{t+T-1} - \bar{\boldsymbol{g}}(\bar{\boldsymbol{\theta}}_t, \zeta_{t+T-1}) + \sum_{k=t}^{t+T-2} \left[ \bar{\boldsymbol{g}}(\bar{\boldsymbol{\theta}}_k, \zeta_k) - \bar{\boldsymbol{g}}(\bar{\boldsymbol{\theta}}_t, \zeta_k) \right] \right\|$$
$$\leq \left\| \bar{\boldsymbol{g}}(\bar{\boldsymbol{\theta}}_{t+T-1}, \zeta_{t+T-1}) - \bar{\boldsymbol{g}}(\bar{\boldsymbol{\theta}}_t, \zeta_{t+T-1}) \right\| + \left\| \mathcal{Y}(\bar{\boldsymbol{\theta}}_t, T-1) \right\|$$
$$= \left\| \boldsymbol{A}_{t+T-1}\left[ \bar{\boldsymbol{\theta}}_{t+T-1} - \bar{\boldsymbol{\theta}}_t \right] \right\| + \mathcal{Y}(\bar{\boldsymbol{\theta}}_t, T-1)$$
$$\leq (1+\gamma) \left\| \bar{\boldsymbol{\theta}}_{t+T-1} - \bar{\boldsymbol{\theta}}_t \right\| + \mathcal{Y}(\bar{\boldsymbol{\theta}}_t, T-1) \tag{92}$$
$$= \alpha(1+\gamma) \left\| \sum_{k=t}^{t+T-2} \left[ \bar{\boldsymbol{g}}(\bar{\boldsymbol{\theta}}_k, \zeta_k) - \bar{\boldsymbol{g}}(\bar{\boldsymbol{\theta}}_t, \zeta_k) + \bar{\boldsymbol{g}}(\bar{\boldsymbol{\theta}}_t, \zeta_k) \right] \right\| + \mathcal{Y}(\bar{\boldsymbol{\theta}}_t, T-1) \tag{93}$$
$$\leq \alpha(1+\gamma) \left[ \mathcal{Y}(\bar{\boldsymbol{\theta}}_t, T-1) + \sum_{k=t}^{t+T-2} \left\| \bar{\boldsymbol{g}}(\bar{\boldsymbol{\theta}}_t, \zeta_k) \right\| \right] + \mathcal{Y}(\bar{\boldsymbol{\theta}}_t, T-1) \tag{94}$$
$$\leq \left[ \alpha(1+\gamma) + 1 \right] \mathcal{Y}(\bar{\boldsymbol{\theta}}_t, T-1) + \alpha(1+\gamma) \left\| \sum_{k=t}^{t+T-2} \bar{\boldsymbol{g}}(\bar{\boldsymbol{\theta}}_t, \zeta_k) \right\| \tag{95}$$

where (92) uses the fact that the spectral radius of $\rho(\boldsymbol{A}(\zeta_t)) \leq 1 + \gamma$ for all $t \in \mathbb{N}^+$; (93) follows from the update in (12); and (94) from the triangle inequality of 2-norm.

We start off by obtaining an bound for the last summand in (95), as follows

$$\mathcal{G}(\bar{\boldsymbol{\theta}}_t, T-1) := \left\| \sum_{k=t}^{t+T-2} \bar{\boldsymbol{g}}(\bar{\boldsymbol{\theta}}_t, \zeta_k) \right\|$$

$$= N^{-1} \sum_{k=t}^{t+T-2} \left\| \sum_{n=1}^{N} \left[ \boldsymbol{A}(\zeta_k)\boldsymbol{\theta}_t^n + r_k^n \boldsymbol{\phi}(s_k) \right] \right\| \tag{96}$$

$$\leq \frac{T-1}{N} \left\| \boldsymbol{A}(\zeta_k)\boldsymbol{\Theta}_t \right\| + (T-1)r_{\max} \tag{97}$$

$$\leq \frac{(T-1)(1+\gamma)}{N} \left\| \boldsymbol{\Theta}_t \right\| + (T-1)r_{\max} \tag{98}$$

$$\leq \frac{(T-1)(1+\gamma)}{N} \left[ \left\| \boldsymbol{\Theta}_t - \bar{\boldsymbol{\theta}}_t \mathbf{1}^\top \right\| + \sqrt{N} \left\| \bar{\boldsymbol{\theta}}_t - \bar{\boldsymbol{\theta}}^* \right\| + \sqrt{N} \left\| \bar{\boldsymbol{\theta}}^* \right\| \right] + (T-1)r_{\max}$$

where (97) is due to the bounds $0 < r_t^n \leq r_{\max}$ for all $m$ and $\|\boldsymbol{\phi}(s_t)\| \leq 1$ for all $t \in \mathbb{N}^+$.

Plugging the bound in (98) into (95) yields

$$\mathcal{Y}(\bar{\boldsymbol{\theta}}_t, T) \leq c_2 \mathcal{Y}(\bar{\boldsymbol{\theta}}_t, T-1) + c_3(T-1) \tag{99}$$

where the coefficients are given by

$$c_2 := \alpha(1+\gamma) + 1 \tag{100a}$$

$$c_3 := \frac{\alpha(1+\gamma)^2}{N} \left\| \boldsymbol{\Theta}_t - \mathbf{1}\bar{\boldsymbol{\theta}}_t^\top \right\| + \frac{\alpha(1+\gamma)^2}{\sqrt{N}} \left\| \bar{\boldsymbol{\theta}}_t - \bar{\boldsymbol{\theta}}^* \right\| + \frac{\alpha(1+\gamma)^2}{\sqrt{N}} \left\| \bar{\boldsymbol{\theta}}^* \right\| + \alpha(1+\gamma)r_{\max}. \tag{100b}$$

Writing the recursion $\mathcal{Y}(\bar{\boldsymbol{\theta}}_t, k)$ in (99) from $k = T$ down to 0, gives rise to

$$\mathcal{Y}(\bar{\boldsymbol{\theta}}_t, T) \leq c_2^T \mathcal{Y}(\bar{\boldsymbol{\theta}}_t, 0) + c_3 \sum_{k=1}^{T} c_2^{T-1}(T-k)$$

$$= c_3 \frac{c_2^T - Tc_2 + T - 1}{(1-c_2)^2}$$

$$= \frac{[\alpha(1+\gamma)+1]^T - T\alpha(1+\gamma) - 1}{\alpha^2(1+\gamma)^2}$$

$$\times \left[ \frac{\alpha(1+\gamma)^2}{N} \left\| \boldsymbol{\Theta}_t - \mathbf{1}\bar{\boldsymbol{\theta}}_t^\top \right\| + \frac{\alpha(1+\gamma)^2}{\sqrt{N}} \left\| \bar{\boldsymbol{\theta}}_t - \bar{\boldsymbol{\theta}}^* \right\| + \frac{\alpha(1+\gamma)^2}{\sqrt{N}} \left\| \bar{\boldsymbol{\theta}}^* \right\| + \alpha(1+\gamma)r_{\max} \right]$$

$$= \left[ \frac{T(T-1)}{2} + \frac{T(T-1)(T-2)}{6}\alpha'(1+\gamma) \right]$$

$$\times \left[ \frac{\alpha(1+\gamma)^2}{N} \left\| \boldsymbol{\Theta}_t - \mathbf{1}\bar{\boldsymbol{\theta}}_t^\top \right\| + \frac{\alpha(1+\gamma)^2}{\sqrt{N}} \left\| \bar{\boldsymbol{\theta}}_t - \bar{\boldsymbol{\theta}}^* \right\| + \frac{\alpha(1+\gamma)^2}{\sqrt{N}} \left\| \bar{\boldsymbol{\theta}}^* \right\| + \alpha(1+\gamma)r_{\max} \right] \tag{101}$$

$$\leq \alpha(1+\gamma)^2 \left[ \frac{T(T-1)}{2} + \frac{T(T-1)(T-2)}{6}\alpha'(1+\gamma) \right]$$

$$\times \left[ N^{-1} \left\| \boldsymbol{\Theta}_t - \mathbf{1}\bar{\boldsymbol{\theta}}_t^\top \right\| + \frac{1}{\sqrt{N}} \left\| \bar{\boldsymbol{\theta}}_t - \bar{\boldsymbol{\theta}}^* \right\| + \frac{1}{\sqrt{N}} \left\| \bar{\boldsymbol{\theta}}^* \right\| + r_{\max} \right] \tag{102}$$

where the first equality arises from the facts that $\mathcal{Y}(\bar{\boldsymbol{\theta}}_t, 0) = 0$ and the summation of geometric series; the second equality uses the results in 100; and (101) employs the mean value theorem which asserts the existence of some $\alpha' \in [0,1]$; while (102) is due to $1 + \gamma \geq 1$.

Then, the term in (88b) can be bounded as follows

$$\left\langle \bar{\boldsymbol{\theta}}_t - \bar{\boldsymbol{\theta}}^*, \sum_{k=t}^{t+T-1} \left[ \bar{\boldsymbol{g}}(\bar{\boldsymbol{\theta}}_k, \zeta_k) - \bar{\boldsymbol{g}}(\bar{\boldsymbol{\theta}}_t, \zeta_k) \right] \right\rangle \leq \left\| \bar{\boldsymbol{\theta}}_t - \bar{\boldsymbol{\theta}}^* \right\| \left\| \sum_{k=t}^{t+T-1} \left[ \bar{\boldsymbol{g}}(\bar{\boldsymbol{\theta}}_k, \zeta_k) - \bar{\boldsymbol{g}}(\bar{\boldsymbol{\theta}}_t, \zeta_k) \right] \right\| \tag{103}$$

$$\leq \mathcal{Y}(\bar{\boldsymbol{\theta}}_t, T) \left\| \bar{\boldsymbol{\theta}}_t - \bar{\boldsymbol{\theta}}^* \right\|$$

$$= \frac{\alpha(1+\gamma)^2}{N} \left[ \frac{T(T-1)}{2} + \frac{T(T-1)(T-2)}{6}\alpha'(1+\gamma) \right]$$

$$\times \left\{ \|\boldsymbol{\Theta}_t - \mathbf{1}\bar{\boldsymbol{\theta}}_t^\top\| \, \|\bar{\boldsymbol{\theta}}_t - \bar{\boldsymbol{\theta}}^*\| + \sqrt{N} \, \|\bar{\boldsymbol{\theta}}_t - \bar{\boldsymbol{\theta}}^*\|^2 + \left[\sqrt{N}\|\bar{\boldsymbol{\theta}}^*\| + N r_{\max}\right] \|\bar{\boldsymbol{\theta}}_t - \bar{\boldsymbol{\theta}}^*\| \right\}$$

$$\leq \frac{\alpha(1+\gamma)^2}{N}\left[\frac{T(T-1)}{2} + \frac{T(T-1)(T-2)}{6}\alpha'(1+\gamma)\right]$$

$$\times \left\{ \frac{1}{2}\|\boldsymbol{\Theta}_t - \mathbf{1}\bar{\boldsymbol{\theta}}_t^\top\|^2 + \frac{1}{2}\|\bar{\boldsymbol{\theta}}_t - \bar{\boldsymbol{\theta}}^*\|^2 + \sqrt{N}\,\|\bar{\boldsymbol{\theta}}_t - \bar{\boldsymbol{\theta}}^*\|^2 \right.$$

$$\left. + \frac{N}{2}\|\bar{\boldsymbol{\theta}}^*\|^2 + \frac{N}{2}r_{\max}^2 + \|\bar{\boldsymbol{\theta}}_t - \bar{\boldsymbol{\theta}}^*\|^2 \right\} \tag{104}$$

$$= \frac{\alpha(1+\gamma)^2}{N}\left[\frac{T(T-1)}{2} + \frac{T(T-1)(T-2)}{6}\alpha'(1+\gamma)\right]$$

$$\times \left\{ \frac{1}{2}\|\boldsymbol{\Theta}_t - \mathbf{1}\bar{\boldsymbol{\theta}}_t^\top\|^2 + \frac{3+2\sqrt{N}}{2}\|\bar{\boldsymbol{\theta}}_t - \bar{\boldsymbol{\theta}}^*\|^2 + \frac{N}{2}\|\bar{\boldsymbol{\theta}}^*\|^2 + \frac{N}{2}r_{\max}^2 \right\}$$

$$\leq \frac{\alpha T^2(1+\gamma)^2\left[3 + T\alpha'(1+\gamma)\right]}{12N}\left[\|\boldsymbol{\Theta}_t - \mathbf{1}\bar{\boldsymbol{\theta}}_t^\top\|^2 + 5N\|\bar{\boldsymbol{\theta}}_t - \bar{\boldsymbol{\theta}}^*\|^2 + N\|\bar{\boldsymbol{\theta}}^*\|^2 + N r_{\max}^2\right] \tag{105}$$

where (103) employs the bound in (102); (104) uses the inequality $2a(b+c) \leq 2a^2 + b^2 + c^2$; and the last inequality is due to $3 + 2\sqrt{N} \leq 5N$ for any $N \geq 1$.

*Bounding the term in* (88c). It is obvious that

$$\mathbb{E}\left[\left\langle \bar{\boldsymbol{\theta}}_t - \bar{\boldsymbol{\theta}}^*, \sum_{k=t}^{t+T-1} \bar{\boldsymbol{g}}(\bar{\boldsymbol{\theta}}_t, \zeta_k) - T\boldsymbol{g}(\bar{\boldsymbol{\theta}}_t)\right\rangle \Big| \mathcal{F}_t \right]$$

$$= \left\langle \bar{\boldsymbol{\theta}}_t - \bar{\boldsymbol{\theta}}^*, \mathbb{E}\left[\sum_{k=t}^{t+T-1} \bar{\boldsymbol{g}}(\bar{\boldsymbol{\theta}}_t, \zeta_k) - T\boldsymbol{g}(\bar{\boldsymbol{\theta}}_t)\Big|\mathcal{F}_t\right]\right\rangle$$

$$\leq \|\bar{\boldsymbol{\theta}}_t - \bar{\boldsymbol{\theta}}^*\| \left\| \mathbb{E}\left[\sum_{k=t}^{t+T-1} \bar{\boldsymbol{g}}(\bar{\boldsymbol{\theta}}_t, \zeta_k) - T\boldsymbol{g}(\bar{\boldsymbol{\theta}}_t)\Big|\mathcal{F}_t\right]\right\|$$

$$\leq T\sigma(T;t)\|\bar{\boldsymbol{\theta}}_t - \bar{\boldsymbol{\theta}}^*\| \left[\|\bar{\boldsymbol{\theta}}_t - \bar{\boldsymbol{\theta}}^*\| + 1\right] \tag{106}$$

$$\leq T\sigma(T;t)\left[2\|\bar{\boldsymbol{\theta}}_t - \bar{\boldsymbol{\theta}}^*\|^2 + \frac{1}{4}\right] \tag{107}$$

where (106) follows from Lemma 5; and (107) uses the inequality $a(a+b) \leq 2a^2 + \frac{1}{4}b^2$.

*Bounding the term in* (88d). For the last term, we have that

$$\left\|\sum_{k=t}^{t+T-1}\left[\bar{\boldsymbol{g}}(\bar{\boldsymbol{\theta}}_k, \zeta_k) - \bar{\boldsymbol{g}}(\bar{\boldsymbol{\theta}}_t, \zeta_k) + \bar{\boldsymbol{g}}(\bar{\boldsymbol{\theta}}_t, \zeta_k)\right]\right\|^2$$

$$\leq 2\left\|\sum_{k=t}^{t+T-1}\left[\bar{\boldsymbol{g}}(\bar{\boldsymbol{\theta}}_k, \zeta_k) - \bar{\boldsymbol{g}}(\bar{\boldsymbol{\theta}}_t, \zeta_k)\right]\right\|^2 + 2\left\|\sum_{k=t}^{t+T-1}\bar{\boldsymbol{g}}(\bar{\boldsymbol{\theta}}_t, \zeta_k)\right\|^2$$

$$= 2\left[\mathcal{Y}(\bar{\boldsymbol{\theta}}_t, T)\right]^2 + 2\left[\mathcal{G}(\bar{\boldsymbol{\theta}}_t, T)\right]^2 \tag{108}$$

$$\leq 8\alpha^2(1+\gamma)^4\left[\frac{T(T-1)}{2} + \frac{T(T-1)(T-2)}{6}\alpha'(1+\gamma)\right]^2$$

$$\times \left[\frac{1}{N^2}\|\boldsymbol{\Theta}_t - \mathbf{1}\bar{\boldsymbol{\theta}}_t^\top\|^2 + N^{-1}\|\bar{\boldsymbol{\theta}}_t - \bar{\boldsymbol{\theta}}^*\|^2 + \frac{1}{N}\|\bar{\boldsymbol{\theta}}^*\|^2 + r_{\max}^2\right]$$

$$+ \frac{8T^2(1+\gamma)^2}{N^2}\|\boldsymbol{\Theta}_t - \mathbf{1}\bar{\boldsymbol{\theta}}_t^\top\|^2 + \frac{8T^2(1+\gamma)^2}{N}\|\bar{\boldsymbol{\theta}}_t - \bar{\boldsymbol{\theta}}^*\|^2 + \frac{8T^2(1+\gamma)^2}{N}\|\bar{\boldsymbol{\theta}}^*\|^2 + 8T^2 r_{\max}^2 \tag{109}$$

$$\leq \left[\frac{2\alpha^2 T^4(1+\gamma)^4}{9N^2}[3 + T\alpha'(1+\gamma)]^2 + \frac{8T^2(1+\gamma)^2}{N^2}\right]\left[\|\boldsymbol{\Theta}_t - \mathbf{1}\bar{\boldsymbol{\theta}}_t^\top\|^2 + N\|\bar{\boldsymbol{\theta}}_t - \bar{\boldsymbol{\theta}}^*\|^2 + N\|\bar{\boldsymbol{\theta}}^*\|^2\right]$$

$$+ \left[ \frac{2\alpha^2 T^4 (1+\gamma)^4}{9} \left[3 + T\alpha'(1+\gamma)\right]^2 + 8T^2 \right] r_{\max}^2 \tag{110}$$

where (108) follows from the definitions of $\mathcal{Y}(\bar{\boldsymbol{\theta}}_t, T)$ and $\mathcal{G}(\bar{\boldsymbol{\theta}}_t, T)$; (109) relaxes both $T - 1$ and $T - 2$ to $T$; and the last inequality arises from their bounds in (102) and (98).

Substituting the bounds in (89), (105), (107), and (110) into (88), gives rise to the following result

$$\mathbb{E}\left[\|\bar{\boldsymbol{\theta}}_{t+T} - \boldsymbol{\theta}^*\|^2 | \boldsymbol{\Theta}_t\right]$$

$$= \|\bar{\boldsymbol{\theta}}_t - \boldsymbol{\theta}^*\|^2 + 2\alpha T\lambda_1 \|\bar{\boldsymbol{\theta}}_t - \boldsymbol{\theta}^*\|^2 + \frac{\alpha^2 T^2 (1+\gamma)^2 [3 + T\alpha'(1+\gamma)]}{6N}\Big[\|\boldsymbol{\Theta}_t - \mathbf{1}\bar{\boldsymbol{\theta}}_t^\top\|^2$$

$$+ 5N\|\bar{\boldsymbol{\theta}}_t - \boldsymbol{\theta}^*\|^2 + N\|\boldsymbol{\theta}^*\|^2 + Nr_{\max}^2 \Big] + 2\alpha T\sigma(T;t)\Big[2\|\bar{\boldsymbol{\theta}}_t - \boldsymbol{\theta}^*\|^2 + \frac{1}{4}\Big]$$

$$+ \left[\frac{2\alpha^4 T^4 (1+\gamma)^4}{9N^2}[3 + T\alpha'(1+\gamma)]^2 + \frac{8\alpha^2 T^2 (1+\gamma)^2}{N^2}\right]\Big[\|\boldsymbol{\Theta}_t - \mathbf{1}\bar{\boldsymbol{\theta}}_t^\top\|^2 + N\|\bar{\boldsymbol{\theta}}_t - \boldsymbol{\theta}^*\|^2 + N\|\boldsymbol{\theta}^*\|^2\Big]$$

$$+ \left[\frac{2\alpha^4 T^4 (1+\gamma)^4}{9}[3 + T\alpha'(1+\gamma)]^2 + 8\alpha^2 T^2\right] r_{\max}^2$$

$$\leq \Bigg[1 + 2\alpha T\lambda_1 + 4\alpha T\sigma(T;t) + \alpha^2 T^2 (1+\gamma)^2 \left[3 + T\alpha'(1+\gamma)\right] + \frac{2\alpha^4 T^4 (1+\gamma)^4}{9N}[3 + T\alpha'(1+\gamma)]^2$$

$$+ \frac{8\alpha^2 T^2 (1+\gamma)^2}{N}\Bigg] \|\bar{\boldsymbol{\theta}}_t - \boldsymbol{\theta}^*\|^2 + \Bigg[\frac{\alpha^2 T^2 (1+\gamma)^2 \left[3 + T\alpha'(1+\gamma)\right]}{6N} + \frac{2\alpha^4 T^4 (1+\gamma)^4}{9N^2}[3 + T\alpha'(1+\gamma)]^2$$

$$+ \frac{8\alpha^2 T^2 (1+\gamma)^2}{N^2}\Bigg] \|\boldsymbol{\Theta}_t - \mathbf{1}\bar{\boldsymbol{\theta}}_t^\top\|^2 + \Bigg[\frac{\alpha^2 T^2 (1+\gamma)^2 \left[3 + T\alpha'(1+\gamma)\right]}{6} + \frac{2\alpha^4 T^4 (1+\gamma)^4}{9N}[3 + T\alpha'(1+\gamma)]^2$$

$$+ \frac{8\alpha^2 T^2 (1+\gamma)^2}{N}\Bigg] \|\boldsymbol{\theta}^*\|^2 + \frac{\alpha^2 T^2 (1+\gamma)^2 \left[3 + T\alpha'(1+\gamma)\right] r_{\max}^2}{6} + \frac{2\alpha^4 T^4 (1+\gamma)^4}{9}[3 + T\alpha'(1+\gamma)]^2 r_{\max}^2$$

$$+ 8\alpha^2 T^2 r_{\max}^2 + \frac{\alpha T\sigma(T;t)}{2} \tag{111}$$

which concludes the proof of Lemma 6.

# I   Additional Numerical Results

We further compare the convergence performance of DTDT and DTDL on a larger size of network, where all the parameters are the same as the ones used in Section 5 except that the total number of agents is 1000 (10 times larger than the previous case). It can be observed in Figure 2 that DTDT still outperforms DTDL with respect to both the consensus error and TD error. Importantly, we can see that the achieved consensus and TD error are not increased compared with the case where there are 100 agents, further strengthening the tightness of our theoretical results.

(a) Error between DTDT and DTDL      (b) Consensus error      (c) TD error

Figure 2: Convergence behaviors of the DTDT and DTDL algorithms