[Reviews · NeurIPS 2020]

Review 1

Summary and Contributions: The paper considers a method for distributed TD approximation with linear function approximation. Geometric convergence to a neighborhood of the optimal solution is shown, both in the case of i.i.d. and Markovian data.

Strengths: The problem of understanding how reinforcement learning works in the multi-agent setting is interesting. The mathematical analysis is plausible and highly correct.

Weaknesses: The main weakness of this paper is that a very similar problem was considered in reference [31], which has a nearly identical title. The main difference between this paper and [31] seems to be in the method: this paper uses gradient tracking, while [31] does not. Nevertheless, both papers show convergence to a neighborhood of the optimal solution, so the theoretical innovation in this paper is not sufficient for NeurIPS publication. The authors mention that the empirical results of this paper are better, but this paper seems focused on theory with a relatively brief simulation section, so it should be evaluated as a theory paper. Additionally, the results here fall short of what one wants to achieve in this setting, which is convergence to the optimal solution. It seems to make more sense to take a decaying step-size and characterize convergence rate to the optimal solution. Additionally, the bounds seem to allow for the possibility that the distance to the optimal solution is quite high if the matrix A has an eigenvalue that is close to zero (if I'm reading Eq. (28)a-(28)c correctly). **Edit:** Thanks to the authors for addressing my points in the rebuttal. Indeed, the bounds here are better by a factor of N than those in [31]; this was already in the submission, but I somehow glazed over it. This answers my main criticism. Additionally, as the authors point out, the analysis is much better, because it is based on a single Lyapunov function. I took a look at the details here and I agree that the analysis is quite elegant and much better than previous work. I raised my score up to a 7. I'm not totally sure my other concern, that the bounds can be bad if A is has an eigenvalue close to zero, have been answered. After all, c_1 depends on 1/(1-rho_i(alpha-bar)), as is clear from the equation just after Eq. (20). Then Eq. (28b) makes clear that if |lambda_1| is close to zero, so is rho_2, and therefore so is rho_i by definition of rho_i. It is possible I'm missing something, so please include a discussion of this. I have another concern upon rereading the paper. Since I did not bring this up at the previous round, and consequently you have not had a chance to rebut this, this does not influence my evaluation. I do not think Algorithm 1 can possibly work as stated. The reason is that for gradient tracking methods initialization is key. They cannot be initialized at phi(0)=0, g(0)=0. I leave it to you to work out a simple example of a deterministic update (i.e., no randomness) on two agents with with f_1(x) = x^2 and f_2(x)=(x-a)^2; while the minimum will occur at x=a/2, the gradient tracking recursion does not depend on a. The only way the number a can appear in the recursion is through the initial condition. The key for the gradient tracking recursion in the deterministic case is to have the average of phi's be the average of the true gradients, otherwise the intuition is lost. In the deterministic case, we therefore need to initialize phi at the initial gradient. Unfortunately, I think this will mean your final result will change somewhat as the initial condition of your Lyapunov function will be different. Please address this issue in the final version of the paper.

Correctness: While I have not checked every detail, the claims seem correct.

Clarity: Yes.

Relation to Prior Work: Reference [31] should be discussed more extensively.

Reproducibility: Yes

Additional Feedback:


Review 2

Summary and Contributions: This paper studies the multi-agent policy evaluation problem. The decentralized TD tracker (DTDT) method is proposed and analyzed to show convergence (with bias) and is verified by empirical results on random graphs.

Strengths: (1) The problem and setting are interesting and important (multi-agent policy evaluation, decentralized TD update). The motivation and presentation are clear. (2) The proposed method is reasonable and makes sense to me. The DTDT method improves the existing DTDL algorithms [24, 11, 31] by adapting the idea of gradient tracking used in decentralized optimization [28, 43]. The results and proofs are mostly correct. (3) Simulation results verified the proposed method. Fig. 1(b) shows the advantages of the gradient tracking.

Weaknesses: (0) Disclaimer: I am also a reviewer for a concurrent submitted paper (with paper id 4026) "A Unifying Finite-Sample Analysis of Decentralized Q-Learning with Linear Function Approximation". (0a) I think this paper tackles different but related problem with paper#4026, i.e., multi-agent policy evaluation by decentralized TD in this paper, and multi-agent control by decentralized Q-learning in paper#4026. (0b) The key technical results and proofs are related and similar, e.g., gradient tracking (which is the key idea in both papers), multi-step Lyapunov function (same in both papers), and similar results (with constant stepsize, the convergence rate is linear rate plus some constant bias). (0c) I think this is ok, but the technical novelty of this paper is weakened after I was aware of another paper using mostly similar techniques to tackle a highly related problem, which has the same main challenges as claimed in both of the two papers (tracking the gradient locally is explicitly state as the main challenge). (1) What are the results for decreasing stepsize? This seems to be a natural and important problem for this algorithm. Minors: (2) In Fig. 1(a), why do the author plot the error between DTDT and DTDL, rather than the error between the two algorithms and optimal solutions, i.e., || \theta^{DTDT} - \theta^* ||, and || \theta^{DTDL} - \theta^* ||. The later errors seem to make more sense, and the current plot cannot really tell why DTDT is better than DTDL. (3) In Remark 1, it is said "improves upon those reported in [11, Thm. 2] and [31, Prop. 2] by a factor of 1/N". I don't get the " improvement over [11, Thm. 2] by a factor of 1/N". In [11, Thm. 2], the error term \beta_2 and \beta_3 are both in O(1) order (which is the same order as the error term in Thm. 1 here?). So why this is an improvement over [11, Thm. 2]. ------Update------ Thanks for the rebuttal and it clarified most of my concerns. I would suggest that results for decaying step-size (which should be similar to the current analysis for constant step-size) appear as theorems in the paper and the authors can put lemmas and proofs in the appendix.

Correctness: I have checked the claims and the proofs mostly, and they look correct to me. The empirical results are also reasonable.

Clarity: The presentation is clear. The paper is well written.

Relation to Prior Work: The related work are well discussed and acknowledged when the authors present lemmas using existing techniques.

Reproducibility: Yes

Additional Feedback:


Review 3

Summary and Contributions: This work proposes the decentralized TD tracking (DTDT) algorithm which solves the policy evaluation problem for multiple agents. A well-established non-asymptotic convergence analysis is built for both i.i.d. samples and the Markovian samples. For both cases, the DTDT algorithm iterations can exponentially converge to a neighborhood of the optimal point with an asymptotical error of at most \alpha level.

Strengths: The main strength of this work is the algorithm design and corresponding convergence analysis. The non-asymptotic analysis of TD learning is popular recently and a key component of reinforcement learning. This paper focuses on improving the convergence error bounds of TD learning in the decentralized setting. The theoretical analysis shows that the error bound achieved by DTDT is independent on the number of agents for both i.i.d. and Markovian cases, implying that the decentralized processing is the same as the centralized processing in terms of iteration complexity. This bound improves the current state of the art result and turns out the best outcome in this field. The numerical result is also encouraging and consistent with the theoretical analysis. It is shown that DTDT can achieve more accurate solution compared with the classic one.

Weaknesses: The results are based on the linear value function approximation. Is possible to extend it to nonlinear case? And tests with large RL tasks? From eq. 9a and 9b, it seems that the algorithm needs two W for communications. Will this cause a communication bottleneck?

Correctness: To my understanding, the theoretical results are correct and make sense. The TD tracker would take a role of estimating the global error so that’s why DTDT can get rid of the number of agents out of the error bound. Numerical results are run on the simple case which is consistent with the theoretical analysis.

Clarity: Yes, the paper is well written.

Relation to Prior Work: Yes, it is clearly discussed.

Reproducibility: Yes

Additional Feedback: 1. To further understand the dependence of asymptotic error term on the number of agents N, I would like to see the numerical experiments by plotting the relation between the asymptotic errors (called residual errors in this paper) and the number of agents N. Also, it would be better to include the codes in the supplementary. 2. For the numerical experiment on Markovian samples, It would be preferred to generate multiple trajectories and draw the average and several percentiles. 3. Another comment on the experiment: The non-asymptotic upper bounds are built for the square distance between the algorithm iteration and the optimal \theta. It would be more persuading if you compute it explicitly rather than use the TD error.


Review 4

Summary and Contributions: The paper focuses on decentralized policy evaluation in multi-agent Markov decision processes using temporal-difference (TD) methods with linear function approximation. The non-asymptotic properties of approach are investigated for both independent and identically distributed (i.i.d.) as well as Markovian transitions.

Strengths: Developing a faster multi-agent reinforcement learning algorithm for policy evaluation along with providing the rigorous statistical proofs are the main strengths of the work, and the topic is relevant to the NeurIPS community. **Edit:** The paper seems a very strong one and the authors addressed the concerns of reviewers appropriately, and I keep the scores the same, indicating a good submission.

Weaknesses: I am not very familiar with this topic and I cannot criticize the paper in comparison to the state of the art or find out its limitations.

Correctness: The paper seems very strong and the claims and method seem to be correct.

Clarity: The paper is very well written and easy to follow. The descriptions of the methods are as an appropriate level of detail, and it is understandable for a person not familiar with the topic.

Relation to Prior Work: The prior work and the differences of this paper from the previous work are properly discussed and cited in the paper.

Reproducibility: Yes

Additional Feedback: A few grammatical and typo problems: - Introduction: Notation: The "i-the" largest eigenvalue - After Eq. (1): can be computed


Review 5

Summary and Contributions: This paper studies a decentralized TD gradient-tracking-based algorithm with linear function approximation. Finite-time convergence bounds are obtained both for the iid and Markov noise setup.

Strengths: Grammatically and structure-wise, the paper is relatively well written. Also, the results improve upon SOTA in decentralized TD, assuming the authors' remarks following the two main theorems are accurate.

Weaknesses: Despite the above statement, this work still lacks clarity: i) There is a confusing leap from the centralized Sec 2.1 to the decentralized Sec 3.1. I find it hard to understand why the state-space is denoted the same in both formulations, while it seems the latter should be the product-space of the former. I expect this because all agents witness the same state-space, but follow their own local policy which should affect the transition to the next state. So, all of them together transition to the same next state. Also, in Sec 3.1. it is unclear why the average R^n and P are independent of \pi in the notation (typo perhaps?). ii) The results contain multiple long and very involved expressions that are more suitable for the appendix rather than the body. I would have appreciated a more parsable O() notation showing the dependence in the most relevant parameters (stepsize, gamma, #agents, etc). Next, I’m bothered by the predominant and repetitive use of the term ‘gradient’ to describe and non-gradient based process. TD(0) is known to be a stochastic approximation algorithm that is *not* a gradient descent algorithm. I would have also appreciated an explanation on why a gradient-tracking method is beneficial for a non-gradient based algorithm.

Correctness: I haven't verified the proofs in detail, but the claims in the various lemmas and theorems seem reasonable and are in the spirit of what I would've expected given other existing results and their dependence in problem parameters.

Clarity: See response in the weaknesses section. Also, a few more minor comments: line 45: the list of two-timescale references lacks a newer SOTA result: Dalal, G., Szorenyi, B., & Thoppe, G. (2019). A tale of two-timescale reinforcement learning with the tightest finite-time bound. AAAI 2020. line 64: typo — “an suitably” line 181: b notation is defined but never used. I assume it was intended to be plugged into (11).

Relation to Prior Work: In terms of contributions, if I understand correctly, the main selling point of this paper is the elimination of the linear dependence of the non-decaying error term in N, for the Markovian error setup. The authors refer to two baselines: [11] and [31]. I tried verifying this claim. In [11], I saw only i.i.d-based convergence theorem, which doesn’t suffer from such linear dependence (N appears in both the numerator and denominator of (14) and (17)). Next, in [31], I also haven’t found such dependence. There, the relevant eq. is (44), which depends on c_5 and c_8’. Trying to dig into these constants, I couldn’t find the alleged N dependence in the error term preceding \alpha. To clarify, I’m not claiming the authors are wrong in claiming their paper to have such merit. But one should be able to verify it easily. Since the task of verifying it is hard enough for me to fail in doing so in a reasonable time, I would have expected a more serious comparison to prior art with exact references to equations.

Reproducibility: No

Additional Feedback:

[Author Response · NeurIPS 2020]

We thank all reviewers for their careful reading of the paper, thoughtful feedback, and constructive suggestions. We look forward to revising the paper in light of your comments. Each reviewer's major comments are addressed below.

**Reviewer 1**. Thanks for your time and effort devoted to reviewing our submission, as well as for the positive comments on the analysis and exposition. Distinct novelties relative to Ref. [31] are: i) *Algorithm:* The present submission develops a novel decentralized multi-agent policy evaluation algorithm by wedding the merits of classic temporal-difference (TD) learning and contemporary gradient tracking, while Ref. [31] analyzed an *existing* distributed multi-agent TD learning algorithm; ii) *Analysis:* Our analysis is based on a single Lyapunov function unifying consensus and convergence while accounting for gradient tracking, yet [31] follows the standard path of proving error bounds for consensus and convergence separately; and, iii) *Bounds:* Our steady-state error bound does not depend on the number of agents $N$, matching that of centralized TD learning, while the steady error bound of [31] is $N$ times *worse*. In a nutshell, although dealing with the same topic, our submission contributes a novel algorithm, a unifying analysis, and improved error bounds relative to [31]. Following your suggestion, [31] will be discussed more thoroughly in the revised paper.

Step-size. We will respectfully disagree that it "makes more sense to take a decaying step-size." In fact, *constant step-sizes* have been used and analyzed in the *seminal* contributions of TD learning [32, 1, 10, 38], as well as in recent ones [3, 5, 15, 31, 40, 41, 44]. Analyzing TD methods with constant step-sizes is critical, and serves as a first-step to analyze decaying step-sizes too. Yet, our algorithm and analysis can be generalized to accommodate decaying step-sizes. Due to space limitation, the focus of this paper was placed on analysis under *both IID and Markovian data*.

Bounds. In (28a)–(28c), $\lambda_1 < 0$ is the largest eigenvalue of the *negative* definite $\mathbf{A}$. If $|\lambda_1|$ is close to zero, it is clear from line 227 that the steady error bound constant $c_1$ is monotonically increasing in $|\lambda_1|$, so the closer $|\lambda_1|$ is to zero, the smaller the steady error bound is. On the other hand, the eigenvalues of $\mathbf{A}$ can be designed (to some extent) by properly selecting features $\{\phi(s)\}_s$; moreover, the error bound can be made arbitrarily small by taking small enough step-size $\alpha > 0$. With these clarifications, it will be great if the reviewer can upgrade the evaluation of our work.

**Reviewer 2**. Comparison with paper #4026. We appreciate your time and effort put in this review, as well as the constructive feedback on our submission. As correctly pointed out, both papers deal with analysis of (decentralized) RL algorithms. The main novelties of this paper are the decentralized TD tracking technique, its corresponding unifying (consensus and convergence) analysis, as well as the resultant state-of-the-art error bound matching the centralized TD setup (improving upon existing distributed TD learning algorithms). Paper #4026 on the other hand, focuses on analysis of multi-agent control using distributed $Q$-learning when both constant and decaying step-sizes are employed (while gradient tracking is *not* used). Policy evaluation and control are two fundamental RL tasks, each of which presents unique challenges in algorithm design and analysis. We agree with your assessment that the two papers deal with related but different RL problems. Step-size. Please refer to lines 13-17 above.

Why plot error between DTDT and DTDL? This is because the optimum $\boldsymbol{\theta}^*$ is *unknown* in practice (in fact, finding $\boldsymbol{\theta}^*$ would require knowing the distribution of the underlying MDP, that is not available in the RL context). Fig. 1a shows that DTDT and DTDL find the same fixed point in the consensus space; while there is a big difference with respect to consensus and TD errors as depicted in Figs. 1b and 1c, corroborating the merits of TD tracking.

Improvement over [11, Thm. 2] by $1/N$. This is due to the fact that the constant $L := \sum_{v \in \mathcal{V}} L_v$ in $\beta_2$ and $\beta_3$ in [11, Thm. 2] is on the order of $N$ (number of agents), as you can see from the sentence above Eq. (40) in the appendix of [11]; while the constants in Thm. 1 of the present submission do not depend on $N$.

**Reviewer 3**. We appreciate your time as well as positive appraisal of this submission. Extension to nonlinear case. There would be some difference regarding the theoretical claims between the linear and nonlinear value function approximation, such as the measure of the optimal solution, but the tracking technique as well as our convergence analysis can be generalized beyond the linear case.

Larger RL tests. As our bounds are (nearly) independent of problem size, the improvement of the proposed DTDT algorithm over DTDL still holds. Following your request, tests with large RL tasks will be added in the revised version.

Two $\mathbf{W}$-communications. Indeed, the improved error bound of DTDT comes at the price of doubling the communication requirements of DTDL. This communication overhead can be challenging in real-time and large-scale RL tasks. Developing communication-efficient alternatives is a fruitful future research direction.

Additional tests. Due to space limitations, only a single test was presented in the original submission. In the revised version, your insightful suggestions for including these additional numerical tests will be accommodated. Codes for the presented experiments are straightforward, so they were not included. However, per your request, codes for all experiments will be provided in the final paper for ease of reproducibility.

**Reviewer 4**. Thanks for the time and effort spent in reviewing this paper, as well as for recognizing its contributions. Thanks also for your careful reading, and pointing out the typos, which will be corrected in the revised version.

[Meta-Review · NeurIPS 2020]

This paper is theoretical work that provides finite time analysis for decentralised TD learning. The reviewers and myself, although not anonymously, think this contribution may be significant and interesting to the community due to recent interest in the finite time analysis of TD algorithms and (linear) function approximation. We request the authors to address the changes required in the manuscript. The authors propose a distributed method for safety. The reviewers and myself were not convinced that this paper proposes a novel method, specifically, due to lack of proper comparison to previous work. I thank both the authors and reviewers for their efforts.